# AVIDa-hIL6: A Large-Scale VHH Dataset Produced from an Immunized Alpaca for Predicting Antigen-Antibody Interactions

**Hirofumi Tsuruta**[1][*], **Hiroyuki Yamazaki**[1][*], **Ryota Maeda**[1][*], **Ryotaro Tamura**[1],
**Jennifer N. Wei**[2], **Zelda Mariet**[2], **Poomarin Phloyphisut**[2], **Hidetoshi Shimokawa**[2],
**Joseph R. Ledsam**[2], **Lucy Colwell**[2], **Akihiro Imura**[1][*]

[1]COGNANO Inc., [2]Google LLC

{tsuruta, yamazaki, maeda, ryotarotamura, akihiroimura}@cognano.co.jp
{weijennifer, zmariet, poomarin, simokawa, jledsam, lcolwell}@google.com

## Abstract

Antibodies have become an important class of therapeutic agents to treat human diseases. To accelerate therapeutic antibody discovery, computational methods, especially machine learning, have attracted considerable interest for predicting specific interactions between antibody candidates and target antigens such as viruses and bacteria. However, the publicly available datasets in existing works have notable limitations, such as small sizes and the lack of non-binding samples and exact amino acid sequences. To overcome these limitations, we have developed AVIDa-hIL6, a large-scale dataset for predicting antigen-antibody interactions in the variable domain of heavy chain of heavy chain antibodies (VHHs), produced from an alpaca immunized with the human interleukin-6 (IL-6) protein, as antigens. By leveraging the simple structure of VHHs, which facilitates identification of full-length amino acid sequences by DNA sequencing technology, AVIDa-hIL6 contains 573,891 antigen-VHH pairs with amino acid sequences. All the antigen-VHH pairs have reliable labels for binding or non-binding, as generated by a novel labeling method. Furthermore, via introduction of artificial mutations, AVIDa-hIL6 contains 30 different mutants in addition to wild-type IL-6 protein. This characteristic provides opportunities to develop machine learning models for predicting changes in antibody binding by antigen mutations. We report experimental benchmark results on AVIDa-hIL6 by using machine learning models. The results indicate that the existing models have potential, but further research is needed to generalize them to predict effective antibodies against unknown mutants. The dataset is available at https://avida-hil6.cognanous.com.

## 1 Introduction

Antibodies are proteins that play an essential role in the immune system. When antigens such as viruses and bacteria invade the body, the immune system protects the body by producing large numbers of antibodies that bind to the antigens to inhibit their function or mark them for removal. Antibodies have become an important class of therapeutic agents to treat human diseases because of their high target specificity and binding affinity [2]. An essential step in therapeutic antibody discovery is the identification of specific interactions between antibody candidates and target antigens, which has traditionally relied heavily on expensive, time-consuming experiments [24]. Therefore,

---

[*]Equal contribution.

37th Conference on Neural Information Processing Systems (NeurIPS 2023) Track on Datasets and Benchmarks.

computational approaches are increasingly used to complement and accelerate traditional processes for therapeutic antibody discovery [46, 25].

In particular, there is growing interest in using machine learning to predict antigen-antibody interactions [39, 28, 20], which can be used to virtually screen binding antibodies against specific target antigens. Schneider *et al.* [39] developed the structure-based deep learning for antibodies virtual screening (DLAB-VS) by using the structural antibody database (SAbDab) [17], which contains collections of antigen-antibody complex structures. Lim *et al.* [28] generated datasets of antibody sequences from mice immunized with cytotoxic T lymphocyte-associated antigen 4 (CTLA-4) and programmed cell death protein 1 (PD-1); then, they built deep learning models to predict binder and non-binder antibodies to CTLA-4 and PD-1. Huang *et al.* [20] proposed AbAgIntPre, a deep learning-assisted prediction method that was trained using only the amino acid sequences in two public databases: SAbDab and the coronavirus antibody database (CoV-AbDab) [35].

Despite these promising developments, progress in therapeutic antibody discovery has lagged behind progress in other areas of drug discovery. A major reason for this is the lack of availability of high-quality, large-scale datasets of antigen-antibody interactions. First, most existing datasets have small sample sizes. For example, as of May 2023, SAbDab and Lim *et al.*'s datasets [28] contain 5,737 and 3,064 binder samples, respectively. In addition, SAbDab only has samples for binding antigen-antibody pairs. In previous studies [39, 20] using SAbDab, antigens and antibodies were randomly paired to form non-binding pairs. CoV-AbDab contains 12,021 entries, from which more than 30,000 antigen-antibody pairs, including non-binding pairs, are available. However, CoV-AbDab provides only the variant name and not the amino acid sequence. As the variant name is defined by representative mutations, the exact amino acid sequence may vary between publications, thus making it difficult to use CoV-AbDab for antibody discovery because a single amino acid change can be critical for an antigen-antibody interaction.

To overcome these limitations, we have developed AVIDa-hIL6, an antigen-variable domain of heavy chain of heavy chain antibody (VHH) interaction dataset produced by an alpaca immunized with the human interleukin-6 (IL-6) protein. IL-6 is a relatively small protein, a simply structured, well-characterized cytokine that exists as a monomer in the body and is associated with many inflammatory diseases and cancers. To ensure a wide variety of antibody sequences, we used VHHs, whose simple structures enable much easier identification of full-length amino acid sequences by DNA sequencing technologies such as next-generation sequencing (NGS) than for conventional antibodies. By leveraging these advantages, AVIDa-hIL6 contains 573,891 antigen-VHH pairs, including 20,980 binding pairs, with their amino acid sequences. In addition, we have developed a novel labeling method to obtain reliable labels for binding and non-binding.

Furthermore, AVIDa-hIL6 contains information on the interaction of diverse VHHs with 30 different mutants produced by artificial point mutations, in addition to the wild-type IL-6 protein. As the COVID-19 pandemic has shown, viruses continuously evolve through mutation to evade the immune system. Because emerging mutations involving amino acid substitutions can lead to profound changes in antibody binding, prediction of their effects is critical in the development of therapeutic antibodies. Notably, AVIDa-hIL6 contains antibody sequences that are positive for most IL-6 mutants but negative for specific IL-6 mutants, or vice versa, thus providing important insights for understanding how antigen mutations affect antibody binding.

The main contributions of this paper are summarized as follows.

- We release AVIDa-hIL6, which is the largest existing dataset for predicting antigen-antibody interactions (10 times larger than any other public dataset) and contains amino acid sequences of antigens and antibodies and binary labels for binding and non-binding pairs.

- AVIDa-hIL6 has the wild type and 30 mutants of the IL-6 protein as antigens, and it includes many sensitive cases in which point mutations in IL-6 enhance or inhibit antibody binding.

- We have designed a novel data generation method, including data labeling, by using the immune system of a live alpaca. This method can be applied to any target antigen, in addition to IL-6.

- We report benchmark results for the prediction of antigen-antibody interactions by using machine-learning-based baseline models. These results confirm that AVIDa-hIL6 provides valuable benchmarks for assessing a model's performance in capturing the impact of antigen mutations on antibody binding.

Table 1: Characteristics of public datasets for predicting antigen-antibody interactions. The numbers of samples were obtained in May 2023. ✓* denotes that relevant information was missing or not available for part of the dataset.

| Dataset | #Samples | | Sequence | | Structure | Antibody type | Data collection |
|---|---|---|---|---|---|---|---|
| | Binder | Non-binder | Antibody | Antigen | | | |
| SAbDab-nano [40] | 1,114 | - | ✓ | ✓ | ✓ | VHH | Curation |
| sdAb-DB [47] | 1,452 | - | ✓ | ✓* | ✓* | VHH | Curation |
| SAbDab [17] | 5,737 | - | ✓ | ✓ | ✓ | Conventional, VHH | Curation |
| Lim *et al.* [28] | 3,064 | 12,055 | ✓ | ✓ | | Conventional | Experiment |
| CoV-AbDab [35] | 32,064 | 5,303 | ✓ | ✓* | ✓* | Conventional, VHH | Curation |
| **AVIDa-hIL6** | 20,980 | 552,911 | ✓ | ✓ | | VHH | Experiment |

## 2 Related Work

In this section, we put our work into context with public datasets for predicting antigen-antibody interactions. Recent advances in NGS technology now enable the construction of large-scale databases of antibody sequences, such as the observed antibody space (OAS) [31] and iReceptor [14]. However, those databases cannot be directly used as training data for predicting antigen-antibody interactions because of the lack of information on the antigen corresponding to each antibody. Thus, those databases are primarily used to build antibody-specific language models [37, 32, 26] and to generate new antibody sequences via deep generative models [3]. Here, we focus on datasets with information on antigen-antibody interactions, as summarized in Table 1.

**Antibody Type.** The datasets listed in Table 1 contain two types of antibodies: conventional and VHH. A conventional antibody comprises two pairs of heavy and light chains. A conventional antibody acts as a single functional unit by combining the heavy and light chains encoded on separate chromosomes. For example, in humans, the heavy chain is encoded on chromosome 14, and the light chain is encoded on chromosome 2 or 22. Therefore, purification of a single-cell lymphocyte is essential for DNA sequencing of the antigen-determining regions on an antibody. Such cell cloning comprises several steps and is time-consuming, making analysis on the order of 10 to the third power or more virtually impossible. In contrast, a VHH, found in camelids such as alpacas and llamas, comprises only heavy chains. The heavy-chain antibodies are derived from a single gene, e.g., they are encoded on chromosome 4 in alpacas; moreover, a VHH is the smallest functional unit of heavy-chain antibodies, and the sequencing for the antigen-determining regions does not require cell cloning. Thus, we can perform exhaustive analysis on the order of six powers of 10 or more by simply extracting DNA from a bulk sample of lymphocytes. Additionally, VHHs have recently gained interest as therapeutic agents because of their small size, high stability, good human tolerability, and relative ease of production [22, 21]. SAbDab-nano [40] and sdAb-DB [47] are public databases that collect only VHHs, but they both have too few samples for machine learning drug discovery or design applications. Hence, we use immunized alpacas as a data source to generate a large amount of VHH sequence data.

**Sequence and Structure Information.** SAbDab [17] and its sub-database, SAbDab-nano [40], collect all the available antigen-antibody complex structures in the Protein Data Bank (PDB) [8]. Also, some data in sdAb-DB [47] and CoV-AbDab [35] include structural information from the PDB. Because accurate knowledge of antibody structures is important for understanding the antigen-binding function of antibodies, SAbDab is increasingly used for antibody structure prediction via deep learning [38, 1]. However, experimental methods for antibody structure determination, such as X-ray crystallography and cryo-electron microscopy, are relatively expensive and time-consuming, making it difficult to increase the amount of data. More recently, machine learning methods such as AlphaFold [23] and RoseTTAFold [6], which accurately predict a protein's structure from the amino acid sequence, have greatly accelerated progress in the biological sciences. AVIDa-hIL6 focuses on amino acid sequences of antigens and antibodies to generate sufficient training data for machine learning.

**Number of Labeled Samples.** Some existing datasets only have samples for binding antigen-antibody pairs. One reason is that the identification of non-binding antigen-antibody pairs has little clinical significance. In previous studies [39, 20] using SAbDab, antigens and antibodies were randomly paired to form non-binding pairs. This process was based on the assumption that

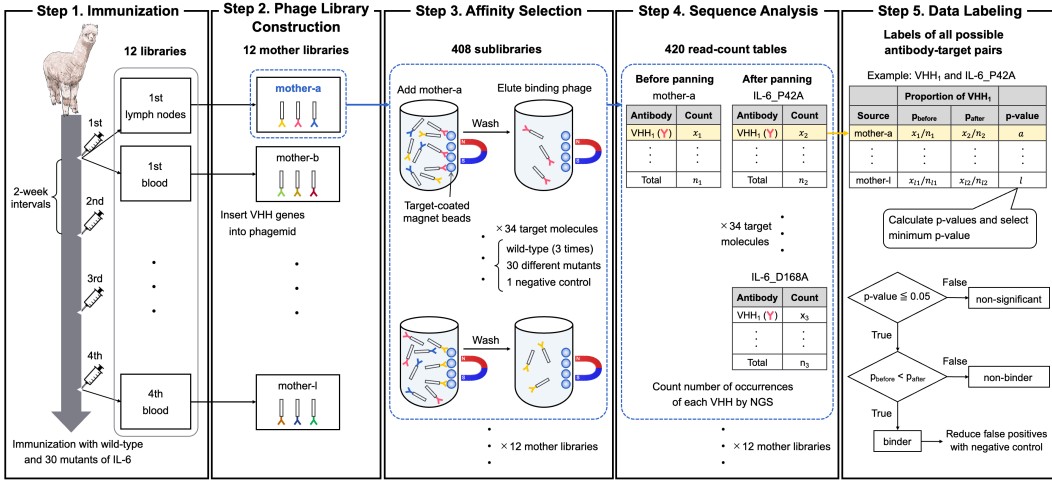

Figure 1: Overview of the data generation process for AVIDa-hIL6.

randomly sampled pairs are unlikely to bind because of antibodies' high target specificity. Lim *et al.* [28] generated a dataset of antibody sequences labeled as "binder" and "non-binder" through an experimental approach using mice immunized with CTLA-4 and PD-1. The number of samples in Table 1 is the total for CTLA-4 and PD-1. As with AVIDa-hIL6, Lim *et al.* created their dataset from their original experiments, thus revealing the potential to increase the amount and diversity of data by the same approach using arbitrary antigens. CoV-AbDab collects antibodies that bind to at least one beta coronavirus and currently contains 12,021 entries. Because each entry has "Binds to" and "Doesn't Bind to" columns and contains zero to multiple antigens for a specific antibody, the number of samples in Table 1 was counted for each possible antigen/antibody pair. However, CoV-AbDab only provides variant names, e.g., SARS-CoV1_Omicron-BA2 or SARS-CoV2_Alpha, and each antigen's exact amino acid sequence is only available if it was provided in the original publication.

# 3 AVIDa-hIL6: Antigen-VHH Interaction Dataset Produced from Alpaca Immunized with Human IL-6 Protein

AVIDa-hIL6 is a dataset of antigen-VHH interactions with amino acid sequences and binary labels for binding and non-binding. In this section, we introduce the dataset generation process, dataset statistics, and verification of label reliability. A labeled dataset and the raw data are available at `https://avida-hil6.cognanous.com`. The dataset is released under a CC BY-NC 4.0 license.

## 3.1 Dataset Generation

Figure 1 shows an overview of the data generation process. The KYODOKEN Institutional Animal Care and Use Committee approved the protocols for the experiments (see Appendix A.1). Appendix A.2 gives the detailed experimental procedures and the amino acid sequences of the IL-6 proteins.

**Step 1. Immunization** To ensure the diversity of antibodies binding to the IL-6 protein that we used as an antigen, we used the immune system of a live alpaca. We introduced a site-directed mutation with alanine at intervals of three to six amino acids, like the alanine scanning technique [15], which is used in molecular biology to determine the contribution of a specific amino acid; as a result, we obtained 30 types of mutants in addition to the wild-type IL-6 protein. A mutant is denoted, for example, as IL6_P42A, which means that an amino acid in the wild type is substituted from proline to alanine at position 42. We immunized a single alpaca with a cocktail of 31 different IL-6 proteins four times at about two-week intervals. After each immunization, one blood sample and one or more lymph nodes from different body sites were collected, yielding a total of 12 libraries. We provide additional information about the collection process at `https://avida-hil6.cognanous.com`.

**Step 2. Phage Library Construction**    We used phage display [43] to identify VHHs that bind to the IL-6 protein. Phage display is a technique for displaying the target proteins on a phage surface in a form that allows them to bind to other molecules. This versatile technique enables generation of protein libraries containing up to $10^{10}$ different variants and is often used for affinity screening of antibodies for their binding partners [7]. We cloned the VHH genes obtained from each library into the pMES4 phagemid vector to display the VHHs on the phagemid surface. As a result, 12 phage libraries corresponding to each of the above libraries were generated and designated as the mother libraries.

**Step 3. Affinity Selection**    Affinity selection by biopanning using the mother libraries can enrich a phage with displayed VHHs that bind to the target molecule. For the target molecules, we used the wild type and 30 mutants of IL-6 and a negative control sample that did not contain any IL-6 protein. Only experiments targeting the wild-type IL-6 protein were performed in triplicate to ensure reproducibility. The mother library was added to the container and incubated with target-coated magnet beads. Then, non-binding phages were washed away, and the remaining phages that bound to the beads were eluted. Consequently, by performing one round of biopanning on each of the 12 mother libraries, we generated a total of 408 sublibraries.

**Step 4. Sequence Analysis**    The amino acid sequences of VHHs displayed on a phage surface can be identified by analyzing the phage genome's DNA with NGS technology. Approximately 100,000 paired reads were generated for each library by NGS, and singletons were removed to avoid sequencing errors. The DNA sequences were translated into amino acid sequences. We counted the number of occurrences of each unique VHH amino acid sequence from the paired reads, which reflected the concentration of each VHH in the library. For each of the 12 mother libraries before panning and 408 sublibraries after panning, we created a table with the VHH amino acid sequences and their read counts.

**Step 5. Data Labeling**    We designed a labeling method to distinguish whether a VHH binds to each IL-6 protein type by applying a statistical test for differences in the proportions of each VHH in a library before and after panning. Here, we focused on examining the binding between a specific VHH and a specific target molecule. Let $p_1$ and $p_2$ denote the population proportions of a specific VHH in the libraries before and after panning. We identified some of the VHH sequences in the libraries by NGS analysis. Let $n_1$ and $n_2$ denote the libraries' total read counts before and after panning, respectively, and let $x_1$ and $x_2$ denote the read counts of a specific VHH in the libraries. Then, the respective sample proportions of a specific VHH in each library are $\hat{p}_1 = \frac{x_1}{n_1}$ and $\hat{p}_2 = \frac{x_2}{n_2}$. Given that the minimum value of all possible $n_1$ and $n_2$ was over 10,000, we assumed that $\hat{p}_1$ and $\hat{p}_2$ follow normal distributions with mean $p_1$ and $p_2$ and variance $\frac{p_1(1-p_1)}{n_1}$ and $\frac{p_2(1-p_2)}{n_2}$, respectively, according to the central limit theorem. Furthermore, the difference in the proportions $\hat{p}_1 - \hat{p}_2$ can also be approximated by a normal distribution due to the reproductive property of the normal distribution. Thus, the test statistic $Z$ under null hypothesis $H_0 : p_1 = p_2$ was calculated as follows.

$$Z = \frac{\hat{p}_1 - \hat{p}_2}{\sqrt{p(1-p)(\frac{1}{n_1} + \frac{1}{n_2})}} \tag{1}$$

where $p$ is the pooled proportion calculated as $p = \frac{x_1+x_2}{n_1+n_2}$. The p-value of $Z$ was calculated using the standard normal distribution. In the same way, p-values were calculated for all VHH-target pairs in the sublibraries with respect to the 12 corresponding mother libraries. Because we had 12 sublibraries associated with the same target molecule, we adopted the smallest p-value, indicating the most significant difference in proportion, among identical VHH-target pairs. If a specific VHH's proportion in a sublibrary increased from the proportion in the corresponding mother library and the p-value was 0.05 or less (our chosen significance level), the VHH-target pair was labeled with "binder." Similarly, if the proportion decreased and the p-value was 0.05 or less, the pair was labeled with "non-binder." Finally, if the p-value exceeded 0.05, the pair was labeled with "non-significant." Our dataset contains 1,998,127 samples labeled non-significant, including 325,865 unique VHH sequences that are present in the alpaca body. These labels were not used for supervised learning to predict antigen-antibody interactions. However, these samples may be helpful for pre-training via self-supervised learning as used by existing antibody-specific language models [37, 32, 26].

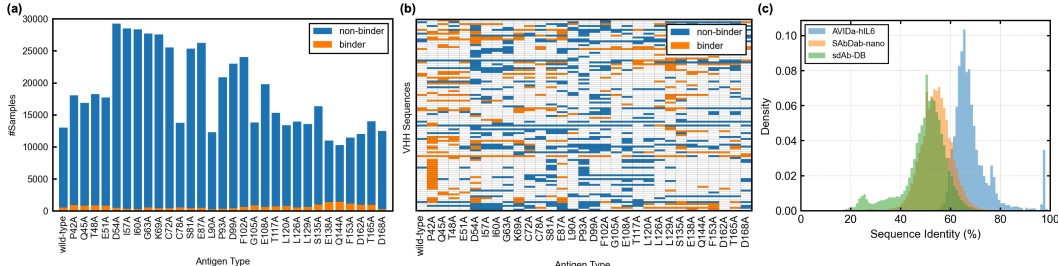

Figure 2: (a) Number of samples for each antigen type. (b) Visualization of the labels for pairs of 100 VHH sequences and each antigen type. Each cell represents a unique VHH-antigen pair. White cells denote non-significant or noise labels. (c) Distribution of the pairwise identities of the VHH sequences.

The results of biological experiments always contain background noise, such as binding to contaminating proteins. Therefore, we developed a novel noise reduction algorithm to avoid false positives and improve label reliability. We reconfirmed VHHs labeled as "binder" to any of the IL-6 proteins by comparing the labels to negative control samples under the following conditions.

1. If the VHH was a non-binder to the negative control sample, the label remained "binder."
2. If the VHH was a binder to the negative control sample, the label was reassigned from "binder" to "noise" because of possible false positives.
3. If the VHH was "non-significant" with respect to the negative control sample, the ratio of the p-value of the negative control sample to that of the IL-6 protein was compared to $10^{2.5}$. This value was empirically determined by an author (a biologist) according to feedback from biological experiments in our previous studies [29].
   (a) If the ratio of p-values was below $10^{2.5}$, the label was reassigned from "binder" to "non-significant" because of possible false positives.
   (b) If the ratio of p-values was $10^{2.5}$ or more, the label remained "binder."

We carefully verified the reliability of our labels, as discussed in section 3.3. The code for data labeling is available at `https://github.com/cognano/AVIDa-hIL6`.

## 3.2 Dataset Statistics

AVIDa-hIL6 contains 573,891 data samples, comprising 20,980 binding pairs and 552,911 non-binding pairs. The proportion of binding pairs is about 3.7 %. Figure 2(a) shows the number of samples for each antigen type. Although the number of samples varied for each IL-6 protein type, we successfully generated at least 10,000 samples for the wild type and 30 different mutants. Furthermore, at least 250 binder VHH sequences existed for each IL-6 protein type. Because we labeled the VHH sequences in the mother library for each IL-6 protein type, AVIDa-hIL6 has information on whether the same VHH sequence binds to each of multiple targets. The number of unique VHH sequences in AVIDa-hIL6 is 38,599, including 4,425 sequences that bind to at least one IL-6 protein type. Importantly, 650 VHH sequences, about 14.7 % of the VHH binders, show binding to specific IL-6 protein types but non-binding to others. We visualized whether 100 sequences extracted randomly from these 650 VHH sequences bound to each antigen type, as shown in Figure 2(b). These samples have valuable information on which mutations enhance or inhibit antibody binding, which should be strongly associated with the IL-6 protein's binding site. Furthermore, when focusing on the same VHH sequence, i.e., one row, we observe that mutants with mutations at closer positions tend to have the same label.

To gain a better understanding of the distribution of VHH sequences, we compared it to the distributions in the existing VHH datasets SAbDab-nano and sdAb-DB. We used only the binders from AVIDa-hIL6 because the existing datasets only contain binders. The numbers of unique VHH binders in SAbDab-nano, sdAb-DB, and AVIDa-hIL6 are 828, 1,414, and 4,425, respectively. To mitigate the computational complexity, we randomly sampled 700 unique VHH sequences from each dataset and calculated all pairwise sequence identities with Biopython v1.81 [13]. Figure 2(c) shows the

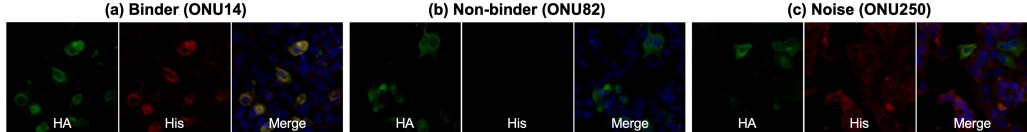

Figure 3: Immunofluorescence staining analysis using three VHHs labeled with (a) binder, (b) non-binder, and (c) noise. HA-tagged IL-6 proteins were introduced into cells and stained with His-tagged VHHs. Subsequently, HA-tags were visualized in green and His-tags in red. The ONU is our identifier to distinguish the VHH sequence.

Table 2: Numbers of samples in the training and test sets.

| Dataset | #Samples | | | Antigen types |
| | Binder | Non-binder | Total | |
| --- | --- | --- | --- | --- |
| Training | 10,564 | 282,279 | 292,843 | Wild-type, Q45A, D54A, G63A, C72A, L90A, P93A, D99A, F102A, G105A, E108A, T117A, S135A, E138A, F153A, D168A |
| Test | 10,416 | 270,632 | 281,048 | P42A, T48A, E51A, I57A, I60A, K69A, C78A, S81A, E87A, L120A, L126A, L129A, Q144A, D162A, T165A |

distributions of sequence identities for these datasets. The results indicate that AVIDa-hIL6 has peaks at regions of higher sequence identity than the other datasets. Interestingly, AVIDa-hIL6 has a peak at 97 % sequence identity, which is absent for the others. As a living organism's immune response progresses, effective antibodies with high binding affinity to the antigen are selected through a process called affinity maturation and are further mutated by repeated exposure to the antigen. Thus, these results may reflect that a live alpaca's immune system selects VHH sequences with high sequence identity that specifically bind to target IL-6 proteins through affinity maturation.

## 3.3 Label Reliability

To verify our label reliability, we tested the antibody binding ability by immunofluorescence staining. Because the number of VHHs that could be verified was limited by the time and cost of biological experiments, VHHs were selected under the following conditions to verify label reliability efficiently. First, we examined only the wild-type IL-6 protein as a target antigen. Next, the amino acid sequences of all labeled VHHs with higher than 93 % identity were clustered by the UCLUST algorithm [18] to validate diverse sequences. When two or more VHHs with the same label were in the same cluster, the one with the highest read count was selected. We know empirically that if all the VHHs in a cluster have the same label, these labels are likely to be true. Therefore, such VHHs were excluded from the candidates. Then, VHHs with suspect labels were selected in order of their read counts. Finally, we tested 10 binder-labeled, six non-binder-labeled, and four noise-labeled VHHs for validation.

Immunofluorescence analysis showed that all 10 binder-labeled VHHs actually bound to the wild-type IL-6 protein, whereas the six non-binder-labeled and four noise-labeled VHHs did not. Figure 3 shows the results for a representative clone of the three types of labeled VHHs. Appendix A.3.2 gives the results for all the tested clones and their amino acid sequences. We could observe the overlapping of IL-6 signals and VHH signals in the binder group, whereas the VHH signals were lost in the non-binder group. The VHH signals did not coincide with the IL-6 signals in the noise group, which can be interpreted as noise-labeled VHHs binding nonspecifically to cells. These results indicate that our noise reduction algorithm contributed to reducing false positives. In addition, these results were also confirmed by kinetic assay via biolayer interferometry (BLI), as described in Appendix A.3.2. As a result, we could ensure that AVIDa-hIL6 has highly reliable labels.

## 4 Benchmarks

### 4.1 Benchmark Task

To demonstrate the use of AVIDa-hIL6 for antibody discovery, we performed an experiment on binary classification of whether a given antigen-antibody pair binds. By leveraging information on the binding of diverse antibodies to antigen mutants, we defined a benchmark task to assess the model performance in capturing the impact of antigen mutations on antibody binding. First, we randomly selected 15 mutants and reserved the data samples for those mutants as a test set. The remaining 15

mutants and the wild type were reserved for model training. Table 2 lists the numbers of samples in the training and test sets. As we used artificial point mutations, each mutant's sequence identity with respect to the wild type was the same, differing only in the position where the alanine was introduced. Next, we trained models by using only the wild-type IL-6 protein and evaluated their performance in predicting antibody bindings in the test set. Then, we randomly selected one mutant from the remaining 15 mutants outside the test set and added it to the training set to evaluate each model's predictive performance. By repeating this process, we tracked the model's predictive performance for unknown mutants contained only in the test set. Because the order of adding mutants to the training set affected the model performance, we ran the same experiment five times in shuffled order, and we report the averaged results here. For all model training, we randomly selected 10 % of the training set for model validation. This experimental scenario assumes that antigen mutants emerge one after another to evade the immune system, as in the COVID-19 pandemic. In such a scenario, we evaluated the model's performance in predicting antibody candidates that will bind to future emerging mutants according to the binding information of antigens that have already been observed.

## 4.2 Baseline Models

We adopted three neural network-based models and one classical machine learning model as baselines. The model inputs were the amino acid sequence of an IL-6 protein with a length of 218 and a VHH with a maximum length of 152.

- **AbAgIntPre** [20] is a state-of-the-art model designed for antigen-antibody interactions based on amino acid sequences. It combines the composition of k-spaced amino acid pairs (CKSAAP) [11] encoding and a convolutional neural network (CNN) model with a Siamese-like architecture. We used the model parameters reported in the original paper [20].

- **PIPR** [10] is a residual recurrent convolutional neural network (RCNN) for protein-protein interaction (PPI) prediction. Following the PIPR strategy, we used an amino acid encoding that combined a five-dimensional vector obtained from a pretrained skip-gram model using the STRING database [44] and a seven-dimensional vector describing the categorization of electrostaticity and hydrophobicity. We changed the number of RCNN units from five to three because our sequence length was more than nine times shorter than the protein input in the original PIPR. The other model parameters were the same as in the paper [10]. Although PIPR was not specifically designed for antigen-antibody interactions, such interactions that ignore non-protein antigens can be considered a subset of PPI, meaning that models designed for PPI can also apply to antigen-antibody interactions.

- **A Multi-Layer Perceptron (MLP)** with one hidden layer of 512 neurons was used as a simpler neural network-based model than the above two models. We used one-hot encoding to represent amino acid sequences. One-hot vectors of the VHHs and IL-6 proteins were flattened and concatenated for input to the MLP.

- **Logistic Regression (LR)** was used as a classical machine learning model that is commonly used for binary classification tasks. As with the MLP, we used flattened and concatenated one-hot vectors of the VHH and IL-6 proteins as input.

In this experiment, the three neural-network-based models were trained for 100 epochs on one NVIDIA Tesla V100 GPU on Google Colaboratory with an initial learning rate of 0.0001 and a batch size of 256. The LR was trained using one Intel(R) Xeon(R) CPU on Google Colaboratory. Appendixes A.4.2 and A.4.3 give more details on the model implementation and training. The code to run the benchmark models is available at `https://github.com/cognano/AVIDa-hIL6`.

## 4.3 Results

Figure 4(a) shows the prediction performance of the baseline models as a function of the number of IL-6 protein types used for model training. We used the precision, recall, and F1-score as evaluation metrics because the prediction of antibody binders, which are fewer in number than non-binders, is much more important for drug discovery. Figure 4(b) shows the precision-recall curves when 1 and 16 IL-6 protein types were used for training. When the number of antigens was 1—that is, when only the wild-type IL-6 protein was used for training—the recalls of AbAgIntPre, PIPR, MLP, and LR were 67.9, 57.6, 67.2, and 67.1 %, respectively. These results indicate that the models failed

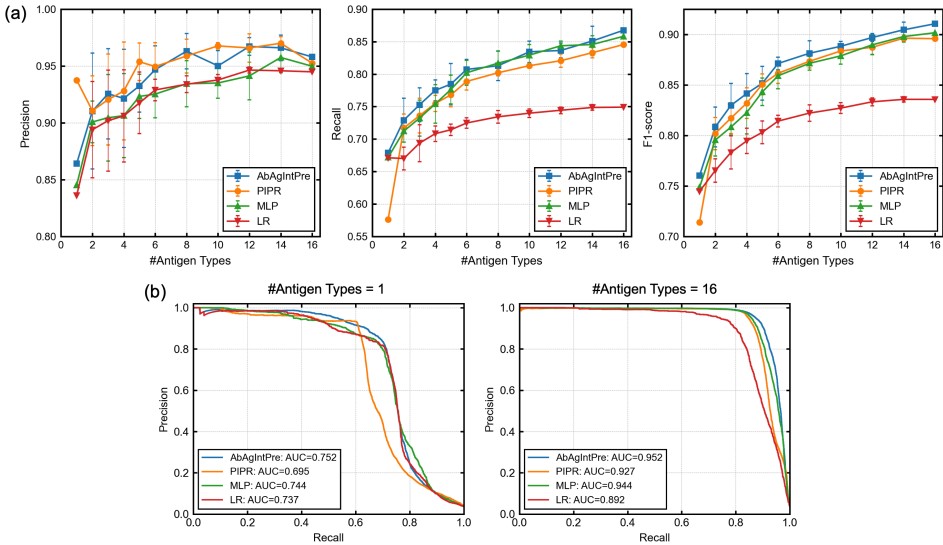

Figure 4: (a) Precision, recall, and F1-score as a function of the number of IL-6 protein types used for model training. (b) Precision-recall curves when 1 and 16 antigen types were used for training. The legend shows the area under the curve (AUC) values.

to predict over 30 % of the effective VHHs that bound to mutants in the test set. All the metrics improved as the number of IL-6 protein types used for training increased, and this trend is clearly evident in the precision-recall curves and the area under the curve (AUC) values. After adding 15 mutants for training, the precisions of all baseline models reached about 95 %, but the recalls were still only about 85 %, even for the three neural-network-based models. A possible factor in this result is that the proportion of binding pairs was only about 3.7 %. Hence, techniques to address imbalanced labels, such as oversampling and undersampling, would be useful to improve the model performance.

For drug discovery applications, the construction of a generalized model for unknown mutations from as little antigen-binding information as possible is ideal, because the number of possible mutations in antigens is tremendously large. As shown by the F1 scores and AUCs in Figures 4(a) and (b), respectively, AbAgIntPre outperformed the other three models, but there was still room for improvement. Furthermore, the performance of AbAgIntPre was not significant as compared to that of the simpler MLP. AVIDa-hIL6 differs significantly from the existing datasets used for training by AbAgIntPre and PIPR because it includes cases in which changes of a few amino acids enhance or inhibit antibody binding. Given this difference in properties, AbAgIntPre and PIPR may not have a clear performance advantage over the MLP. Hence, these results indicate the need for research on model architectures that are dedicated to predicting antibody binding to antigen mutants, and AVIDa-hIL6 will be a useful benchmark for evaluating such models.

## 5 Discussion

### 5.1 Binding Site Prediction

Antibodies recognize specific regions of antigens, called epitopes, and the regions of antibodies that are directly involved in recognition are called paratopes. The antigen-antibody interaction is defined between the epitope and the paratope and is governed by van der Waals forces, electrostatic forces, hydrogen bonding, hydrophobic interactions, and entropic changes at the binding site, which comprise amino acids and their chemical modifications such as phosphorylation, glycosylation, lipidation, methylation, acetylation, or ubiquitylation. Because epitopes and paratopes are crucial for the affinity and specificity of antigen-antibody interactions, many studies have been devoted to predicting epitopes [42, 45] and paratopes [27, 12]. AVIDa-hIL6 has highly sensitive information on changes of a few amino acids in both the antigen and antibody that can significantly affect binding, which should be strongly associated with epitopes and paratopes. Thus, AVIDa-hIL6 may facilitate research on predicting epitopes and paratopes from amino acid sequences.

## 5.2 Potential Risk

In recent years, VHH technology has rapidly developed not only as a research and diagnostic tool but also as a therapeutic agent. VHHs are known to have low toxicity to humans, and several VHH drugs have been approved to date [4]. Antibody genes are activated in B lymphocytes, and their complementarity-determining regions coding paratopes are the only genes in which unrestricted mutations occur *in vivo*. Hence, antibodies that happen to bind to their own tissues can be produced. Mammals have an immune tolerance system that minimizes the production of such autoreactive antibodies through several mechanisms [19]. As our dataset was derived from alpacas, even if the risk of autoimmune adverse events is low for alpacas, it may not be for humans. Therefore, a phase I clinical trial cannot be omitted for each clone for the time being.

## 5.3 Potential Data Biases

Close examination of our data revealed that the probability of the presence of binder VHHs varied dynamically depending on when and where the mother library was collected. Furthermore, because our dataset was produced in an alpaca's body, antibodies that strongly interact with proteins in an alpaca's body were likely to be excluded because they cause autoimmune disease. Therefore, our dataset potentially contains data biases derived from the timing and body site of the library collection and the specific alpaca used in the experiments. To reduce these data biases, it would be beneficial to collect samples at multiple times of immunization, from multiple individuals with different VHH gene sequences, and from multiple animals of different species.

## 5.4 Limitations and Future Works

We introduce two potential limitations of AVIDa-hIL6 and describe future works to address them. The first limitation is that AVIDa-hIL6 uses artificial mutations. Such mutations offer the advantage of investigating binding to an arbitrary number of mutants; however, natural mutations are more complex, as different sites mutate simultaneously. Furthermore, natural mutants include those that have similar functions and structures even with different amino acid sequences, and point mutations that cause loss or gain of the antigenic protein's function. Because simple artificial mutants have little chance of reproducing these rare properties, the efficiency of data collection for predicting antigen-antibody interactions of mutants with these properties is low. The second limitation is the lack of antigen diversity: specifically, AVIDa-hIL6 only has the IL-6 protein as an antigen. Our experimental scenario is to predict antibody binding to unknown mutants of a known antigen. In drug discovery applications, there is also a need to find effective antibodies against new emerging antigens. These limitations lead to the narrow applicability of a model trained on AVIDa-hIL6.

An essential approach to overcome these limitations will be to accumulate labeled data for a wider variety of antigens and their mutants. Because our data generation method described in section 3.1 is applicable to any target antigen, it can be a fundamental technology for establishing a more comprehensive database of antigen-antibody interactions. In fact, we used the same approach to generate a dataset for SARS-CoV-2 variants and successfully found effective antibodies [29]. In the future, we plan to generate and release datasets for various antigens, which should be more practical for building models to predict antigen-antibody interactions. Furthermore, we will explore combining AVIDa-hIL6 with other data sources, such as those listed in Table 1.

## 6 Conclusion

In this paper, we have described AVIDa-hIL6, a large-scale dataset of IL-6 protein-VHH pairs containing amino acid sequence information and reliable labels for binding or non-binding pairs. By introducing artificial mutations into the IL-6 protein used as an antigen, we generated an interaction dataset for 30 types of mutants in addition to wild-type IL-6. This design enabled AVIDa-hIL6 to include many sensitive cases in which point mutations in the IL-6 protein enhance or inhibit antibody binding, thus providing researchers with valuable insights into the effects of antigen mutations on antibody binding. We envision that AVIDa-hIL6 will help democratize antibody discovery and serve as a valuable benchmark for machine learning research in the growing field of predicting antigen-antibody interactions.

## Acknowledgments and Disclosure of Funding

We thank Tomohisa Oda for developing the AVIDa-hIL6 website.

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

# A    Appendix

## A.1    Ethics Statement for Animal Experiments

All animal experiments on an alpaca were conducted in accordance with the KYODOKEN Institute for Animal Science Research and Development (Kyoto, Japan) and the ARRIVE (Animal Research: Reporting of *In Vivo* Experiments) guidelines[2]. Veterinarians performed breeding, health maintenance, and immunization by adhering to the published Guidelines for Proper Conduct of Animal Experiments by the Science Council of Japan. The KYODOKEN Institutional Animal Care and Use Committee approved the protocols for these studies (KYODOKEN protocol number 20190216).

Our data generation method uses animal models immunized with a target protein that could potentially harm the animals, such as a particular toxin, pathogen, or allergen. Hence, the risk to the animal should be minimized by treating the immune source to inactivate or detoxify it.

## A.2    Dataset Generation

Here, we describe the detailed experimental procedures and conditions for each step.

**Step 1.    Immunization**    We immunized a single alpaca with purified recombinant human IL-6 protein and several single-amino-acid mutants.    Specifically, the gene encoding the human IL-6 protein was codon-optimized, synthesized, and sub-cloned in the pcDNA3.1(+) vector (Thermo Fisher Scientific K.K., Tokyo, Japan).    The amino acid sequence of the wild-type IL-6 protein with a C-terminal $6\times$His-tag was "MNSFSTSAFGPVAFSLGLLLVLPAAFPAPVPPGEDSKDVAAPHRQPLTSSERIDKQIRY-ILDGISALRKETCNKSNMCESSKEALAENNLNLPKMAEKDGCFQSGFNEETCLVKIITGL-LEFEVYLEYLQNRFESSEEQARAVQMSTKVLIQFLQKKAKNLDAITTPDPTTNASLLTKL-QAQNQWLQDMTTHLILRSFKEFLQSSLRALRQMHHHHHH." We introduced a site-directed mutation with alanine at intervals of three to six amino acids, like the alanine scanning technique [15], which is used in molecular biology to determine the contribution of a specific amino acid. A total of 30 single-amino-acid mutants was prepared: P42A, Q45A, T48A, E51A, D54A, I57A, I60A, G63A, K69A, C72A, C78A, S81A, E87A, L90A, P93A, D99A, F102A, G105A, E108A, T117A, L120A, L126A, L129A, S135A, E138A, Q144A, F153A, D162A, T165A, and D168A. Here, for example, P42A means that an amino acid in the wild type was substituted from proline to alanine at position 42. The antigen cocktail mixture was emulsified in Titermax (Funakoshi, Tokyo, Japan) adjuvant at a dose of 600 $\mu$g and subcutaneously injected into an alpaca four times at about two-week intervals. Lymph nodes and blood samples were each collected four times, resulting in a total of 12 libraries.

**Step 2. Phage Library Construction**    Peripheral blood mononuclear cells (PBMCs) were obtained from blood samples by sucrose density gradient centrifugation using Ficoll (Nacalai Tesque, Kyoto, Japan). The lymph nodes and PBMC samples were washed with phosphate-buffered saline (PBS, Nacalai Tesque) and suspended in an RNAlater solution (Thermo Fisher Scientific K.K., Tokyo, Japan). Total RNA was isolated from these samples by using Direct-Zol RNA MiniPrep (Zymo Research, Irvine, CA). Complementary DNA was synthesized from 1 $\mu$g of total RNA as a template by using random hexamer primers and SuperScript II reverse transcriptase (Thermo Fisher Scientific K.K.). The coding regions of the VHH domain were amplified using LA Taq polymerase (TAKARA Bio Inc., Shiga, Japan) with two PAGE-purified primers (CALL001, 5'-GTCCTGGCTGCTCTTCTACAAGG-3' and CALL002, 5'-GGTACGTGCTGTTGAACTGTTCC-3'), and they were separated on a 1.5 % low-melting-temperature agarose gel (Lonza Group AG, Basel, Switzerland). Approximately 700 base-pair bands were extracted using a QIAquick Gel Extraction Kit (Qiagen K.K., Tokyo, Japan). Nested PCR was performed to amplify the VHH genes by using two primers that contained flanking PstI (forward) and BstEII (reverse) restriction sites to enable cloning into the pMES4 phagemid vector with a C-terminal His-tag. Electroporation-competent Escherichia coli TG1 cells (Agilent Technologies Japan, Ltd., Tokyo, Japan) were transformed with the ligated plasmids under chilled conditions (Bio-Rad Laboratories, Inc., Hercules, CA). The library densities were monitored and maintained at $>10^7$ colony-forming units per microliter with limiting dilution. Colonies from 8 mL of cultured cells were harvested, pooled, and reserved in frozen glycerol stock as a mother library. Thus, the 12 phagemid libraries were designated as the mother libraries.

---

[2]ARRIVE guidelines: `https://arriveguidelines.org`

**Step 3. Affinity Selection**   One round of biopanning was performed using each target protein-coated magnet beads in 50-mM phosphate buffer (pH 7.4) containing 0.1 % Triton X-100 (Nacalai Tesque), 0.3 % (w/v) bovine serum albumin (BSA, Nacalai Tesque), and 500 mM of NaCl. Every IL-6 mutant was used at 1.2 mL bead slurry, which was saturated with 240 $\mu$g of protein, except for P93A (90 $\mu$g), E108A (190 $\mu$g), and L126A (180 $\mu$g). To distinguish nonspecific signals, a negative control sample that did not contain any IL-6 protein was also used. The wild-type IL-6 protein libraries were obtained in triplicate to confirm the reproducibility. After three washes with the same buffer, the remaining phages bound to the beads were eluted with a trypsin-ethylenediaminetetraacetic acid (EDTA, Nacalai Tesque) solution at room temperature for 30 minutes. The eluate was neutralized with a PBS-diluted protein inhibitor cocktail (cOmplete, EDTA-free, protease inhibitor cocktail tablets, Roche Diagnostics GmbH, Mannheim, Germany) and used to infect electroporation-competent cells. The infected cells were cultured in LB Miller broth containing 100 $\mu$g/mL of ampicillin (Nacalai Tesque) at 37 °C overnight. The genes of the phagemids selected by biopanning were collected with a QIAprep Miniprep Kit (Qiagen), amplified by PCR, and purified using AMPure XP beads (Beckman Coulter, High Wycombe, UK). Then, dual-indexed libraries were prepared and sequenced on an Illumina MiSeq (Illumina, San Diego, CA) by using a MiSeq Reagent Kit v3 with paired-end 300-bp reads (Bioengineering Lab. Co., Ltd., Kanagawa, Japan).

**Step 4. Sequence Analysis**   Approximately 100,000 paired reads for each library were generated by NGS analysis. The raw read data were trimmed to remove the adaptor sequence by using cutadapt v1.18 [30] and to remove low-quality reads by using Trimmomatic v0.39 [9]. The remaining paired reads were merged using fastq-join [5], and then the VHH coding sequences were extracted using seqkit v0.10.1 [41]. The DNA sequences were translated to amino acid sequences with EMBOSS v6.6.0.0 [36], and the VHH sequences were cropped from start to stop codon. Finally, each phagemid library was converted to a FASTA file containing tens of thousands of VHH sequences.

## A.3   Label Reliability

### A.3.1   Experimental Procedures

**VHH Substantiation**   The gene sequences encoding each selected VHH clone, which were connected with a 4×(GGGGS) linker for expression as a tandem dimer, were codon-optimized and synthesized (Eurofins Genomics Inc., Tokyo, Japan). The synthesized genes were subcloned into the pMES4 vector to express N-terminal PelB signal peptide-conjugated and C-terminal 6×His-tagged VHHs. BL21 (DE3) E. coli cells transformed with the plasmids were plated on LB agar with ampicillin and incubated at 37 °C overnight. Grown colonies were picked and cultured at 37 °C to reach an OD of 0.6 AU, and the cells were then cultured at 37 °C for three hours with 1 mM of IPTG (isopropyl-$\beta$-D-thiogalactopyranoside, Nacalai Tesque). Lastly, the cultured cells were pelleted by centrifugation and stored in a freezer until use. VHHs were eluted from the periplasm by soaking in TES buffer (200 mM Tris, 0.125 mM EDTA, 125 mM sucrose, and pH 8.0) at 4 °C for one hour. They were further incubated with a 2× volume of 0.25× diluted TES buffer with a trace amount of benzonase nuclease (Merck) at 4 °C for 45 minutes. The supernatants were centrifuged (20,000 ×g, 4 °C for 10 minutes), sterilized by adding gentamicin (Thermo), and passed through a 0.22 $\mu$m filter (Sartorius AG, Gottingen, Germany). The filtered supernatants were then applied to a HisTrap HP nickel column (Cytiva) on an ÄKTA pure HPLC system, and the bound His-tagged VHHs were eluted with 300 mM of imidazole. The eluted fraction was collected and concentrated with a VIVAspin 3000-molecular-weight cutoff filter column (Sartorius) and applied to a Superdex75 10/300 GL gel-filtration column (Cytiva) on an ÄKTA pure HPLC system. Finally, the protein purity was measured via Coomassie brilliant blue (CBB) staining (Rapid Stain CBB Kit, Nacalai Tesque).

**Immunofluorescence Staining Analysis**   HEK293T cells were transiently transfected with a plasmid-encoding C-terminally HA-tagged wild-type IL-6 protein by using Lipofectamine 3000 (Thermo) according to the manufacturer's instructions. The next day, the cells were seeded on collagen type-I-coated culture plates (IWAKI, AGC TECHNO GLASS CO., LTD., Shizuoka, Japan) and cultured for 24 hours before being fixed with 2 % paraformaldehyde (PFA) at 4 °C overnight. After three washes with PBST (PBS with 0.005 % Tween 20), the cells were blocked with PBST containing 2 % goat serum (blocking solution) at room temperature for one hour. Each well was soaked with 100 $\mu$L of the blocking solution containing 100 ng of purified VHH at 4 °C overnight. After washing with PBST, 1:3000-diluted anti-His-tag rabbit antibodies and 1:100-diluted anti-HA 7C9 mouse

Table 3: Amino acid sequences of the VHHs used for label verification.

| VHH | Label | Amino acid sequence |
|---|---|---|
| ONU7 | binder | MKYLLPTAAAGLLLLAAQPAMAQVQLQESGGGLVQPGGSLRLSCAASGFTFSSYAMSWVRRAPGKGLEWVSHISTSGG FTTYLDSVKGRFTISRDNAKNMLYLQMSSLKPEDTAVYYCAESRGMVGASYAAYVDKGTQVTVSSHHHHHH |
| ONU14 | binder | MKYLLPTAAAGLLLLAAQPAMAQVQLQESGGGLVQPGGSLRLSCAASGIASINALGYYRQAPGKQRELVAAVTGGGRT NYADSVKGRFTISRDNAKNTVYLQMNSLKPEDTAVYYCNAKRWGSDYWGQGTQVTVSSHHHHHH |
| ONU54 | binder | MKYLLPTAAAGLLLLAAQPAMAQVQLQESGGGLVQPGGSLRLSCVTSGFTSDYYAIGWFRQAPGKAREGVSCISSSGG GVDYEDSVKGRFTISRDNAENTVHLQMNSLKPEDTAVYYCAAYRSKYGCSRDLRLYDYWGQGTQVTVSSHHHHHH |
| ONU65 | binder | MKYLLPTAAAGLLLLAAQPAMAQVQLQESGGGLVQAGGSLRLSCAASGSSESNYAMGWFRQAPGKEREFVAAISWSGG STYYADSVKGRFTISRGNAKNTVYLQMNSLKPEDTAVYYCAAKPIAYYNDEYEYWGQGTQVTVSSHHHHHH |
| ONU88 | binder | MKYLLPTAAAGLLLLAAQPAMAQVQLQESGGGLTQPGGSLRLSCAASGNSRSINAMGWSRQAPGKQRDLVALITSGGT TAYGESVKGRFTISRDNADNTVWLQMNSLKPEDTAVYYCYAVSDGNSRQYWGQGTQVTVSSHHHHHH |
| ONU90 | binder | MKYLLPTAAAGLLLLAAQPAMAQVQLQESGGGLVQAGGSLRLSCAASGLTFTRYHMAWFRQAPGKEREMVAAISWSGS TTDYQDSVKGRFTISRDNAKNTVSLQMNNLKPDDTAVYYCAASQTRALAPLIGRYDYWGQGTQVTVSSHHHHHH |
| ONU174 | binder | MKYLLPTAAAGLLLLAAQPAMAQVQLQESGGGLVEPGGSLRLSCAASKFTLAYYDIAWFRQAPGKEREGVSCISSYDG STYYADSVKGRFTISRDNAKNTVYLQMNSLKPEDTAIYFCATDHTGAPKCSMKTIGEYNYRGQGTQVTVSSHHHHHH |
| ONU191 | binder | MKYLLPTAAAGLLLLAAQPAMAQVQLQESGGGLVQPGGSLRLSCVASGFTSDPYAIGWFRQAPGKEREGVSCISSSGG SIEYEDSVKGRFTISRDNAENTVHLQMNSLKPEDTAVYYCAAYRSKYGCARELDLYDYWGQGTQVTVSSHHHHHH |
| ONU290 | binder | MKYLLPTAAAGLLLLAAQPAMAQVQLQESGGGLGQAGGSLTLSCAASEGIGSVNAMGWYRQAPGKQRELVAAISRGGG IMYADSVKGRFTISRDNAKNTVYLQMNSLKPEDTAVYYCAADRVILLFDSRSADYWGQGTQVTVSSHHHHHH |
| ONU455 | binder | MKYLLPTAAAGLLLLAAQPAMAQVQLQESGGGLVQAGGSLRLSCAASGTIFTINTMGWYRQAPGKQRELVASITSDGS TNYANSLKGRFTISRDNAKNTVYLQMNSLKPEDTAVYYCAAGWYDRGDDYWGQGTQVTVSSHHHHHH |
| ONU1160 | binder | MKYLLPTAAAGLLLLAAQPAMAQVQLQESGGGLVQPGGSLRLSCTASGFTLDDYAIGWFRQGPGKEREGVSCISSSDG STYYLDSVKGRFTISRDNAKNTVYLSMNSLNVEDTGVYYCAADRSCWAYMDYWGKGTQVTVSSHHHHHH |
| ONU1881 | binder | MKYLLPTAAAGLLLLAAQPAMAQVQLQESGGGLVQPGGSLRLSCAASRFTLAYYDIGWFRQAPGKEREGVSCISSYDG STYYADSVKGRFTISRDNAKNTVYLQMNSLKPEDTAIYYCATDHTGAPTCSTKSIGQYDYRGQGTQVTVSSHHHHHH |
| ONU57 | non-binder | MKYLLPTAAAGLLLLAAQPAMAQVQLQESGGGLVQPGGSLTLACAASGSILDIDIMRWYRQAPGEQREIVATITNSGT TTYRDSVKGRFTISRDTAENTVYLQMNSLKPEDTAVYTCQADVYVNGDDDKFQFFGFWGQGTQVTVSSHHHHHH |
| ONU60 | non-binder | MKYLLPTAAAGLLLLAAQPAMAQVQLQESGGGLVQPGGSLRLSCAASGFTLDVYAIAWFRQAPGKEREWVSCISESVG ATLYAESVKGRFTISRDNAKNTVYLQMNSLKPEDTAVYYCAPPLECSGYGLTKLHDSRSQGTQVTVSSHHHHHH |
| ONU82 | non-binder | MKYLLPTAAAGLLLLAAQPAMAQVQLQESGGGLVQPGGSLRLSCAASGRMGNINVLGWYRQAPEKQRELVATITNFGT IKYGDSVKGRFIISKNSAWNMVYLQMNSLKPEDTAVYYCNAANRIGPEKKMDDYWGQGTQVTVSSHHHHHH |
| ONU84 | non-binder | MKYLLPTAAAGLLLLAAQPAMAQVQLQESGGGLVQAGDSLRLSCAASGGIFSRYAMGWFRQAPGKEREIVAAISWSGG STRYGDSVKGRFTISRDNAKNTVYLQMNSLKPEDTAVYYCAATISPTYYTGTYAYTSTYDDWGQGTQVTVSSHHHHHH |
| ONU171 | non-binder | MKYLLPTAAAGLLLLAAQPAMAQVQLQESGGGLVQAGGSLRLSCAASGFAFGDYAIGWFRQAPGKEREAVSCISNTDG ITHYVDSVKGRFTISSDNAKNTVYLQMSSLKPEDTAVYYCAASSQGSGYHYCSGSAYIRVGMDWWGKGTQVTVSSHHHHHHHH |
| ONU232 | non-binder | MKYLLPTAAAGLLLLAAQPAMAQVQLQESGGGLVQPGGSLRLSCTASGLTFSIYAMSWVRQAPGKGLEWVSDINSDGD NAYYADSVKGRFTISRDNAKNTVDLQMNSLKPEDTGVYYCATDRRSTIARMVRRTDFGSWGQGTQVTVSSHHHHHH |
| ONU2 | noise | MKYLLPTAAAGLLLLAAQPAMAQVQLQESGGGLVQPGESLRLSCAASGRTDSRYAVAWFRQAPGKARELVSSISWDAG LTHYADFVKGRFAISRDNAKNMVYLQMNSLEFEDTAVYYCAAAYYDGSRLFKVIYDYWGQGTQVTVSSHHHHHH |
| ONU3 | noise | MKYLLPTAAAGLLLLAAQPAMAQVQLQESGGGLVQAGGSLRLSCIASGSTFSSYRMGWFRQAPGKEREFVAAISHFGI STYYADSVKGRFTISRDNAKNIVYLQMNSLKPEDTASYYCAADGDPYHRNYERLGEYDYWGQGTQVTVSSHHHHHH |
| ONU4 | noise | MKYLLPTAAAGLLLLAAQPAMAQVQLQESGGGLVQSGGSLRLSCAASGFSLDYYNIGWFRQAPDKDREGVSCISSSGS STNYADSVKGRFTISRDNAKNTVYLQMNSLKPEDTAVYYCVVDDGRVGCTEARRTLAYDYWGQGTQVTVSSHHHHHH |
| ONU5 | noise | MKYLLPTAAAGLLLLAAQPAMAQVQLQESGGGLVQAGGSLTLSCAASGRTFSTDAMGWFRQAPGKEREFVATVSWGGG NTYYADTVKGRFTIFRDNAKNTVYLQMNNLEPEDTAVYYCATSLTTTHMRAREVDYWGQGTQVTVSSHHHHHH |
| ONU250 | noise | MKYLLPTAAAGLLLLAAQPAMAQVQLQESGGGLVQSGGSLRLSCAASGFSLDYYNIGWFRQAPDKDREGVSCISSSAS SSTNYADPVKGRFTISRDNAKNTVYLQMNSLKPEDTAIYYCVVDDGRVGCTEARRTLAYDYWGQGTQVTVSSHHHHHH |

monoclonal antibodies (ChromoTek GmbH, Planegg-Martinsried, Germany) in blocking buffer were added and reacted at room temperature for one hour. Finally, after washing, Alexa-Fluor-conjugated anti-rabbit IgG (594 nm emission) antibodies at 1:3000 dilution and Alexa-Fluor-conjugated anti-mouse IgG (488 nm emission) antibodies at 1:3000 dilution in blocking buffer were added to the wells, and the fixed cells were labeled at room temperature for one hour before washing three times with PBST. The cell nuclei were visualized with 4',6-diamidino-2-phenylindole (DAPI). The stained cells were imaged with an 8-ms exposure time (594 nm emission), a 40-ms exposure time (488 nm emission), or an automatically adjusted exposure time (DAPI) by using an IX71S1F-3 microscope (Olympus Corporation, Tokyo, Japan) with the cellSens Standard 1.11 application (Olympus). Each full observed field corresponding to a 165 $\mu$m $\times$ 220 $\mu$m square was photographed.

**Kinetic Assays via Biolayer Interferometry (BLI)**    Real-time binding experiments were performed using an Octet Red96 instrument (fortèBIO, Pall Life Science, Portsmouth, NH). Each purified VHH clone was biotinylated with EZ-Link Sulfo-NHS-LC-Biotin (Thermo) according to the manufacturer's protocol; uncoupled biotin was excluded with a size exclusion spin column (PD SpinTrap G-25, Cytiva) in PBS (pH 7.4). Assays were performed at 30 °C with shaking at 1000 rpm. Biotin-conjugated clones at 10 $\mu$g/mL were captured on a streptavidin-coated sensor chip (SA, fortèBIO) to reach the signals at 1 nm. One unrelated VHH P17-coated sensor chip was monitored as a baseline. The loaded concentration of the wild-type IL-6 was 200 $\mu$g, corresponding to 0.625 $\mu$M. Assays were performed with PBS containing 0.005 % Tween 20 (Nacalai Tesque). After baseline equilibration for 180 s in the buffer, association and dissociation were each performed for 180 s. The data were then subtracted from the baseline data and analyzed with fortèBIO data analysis software 9.0.

### A.3.2    Additional Results

As listed in Table 3, we selected 12 binder-labeled, six non-binder-labeled, and five noise-labeled clones for substantiation. Of the 12 binder-labeled clones, two could not be isolated by the E. coli

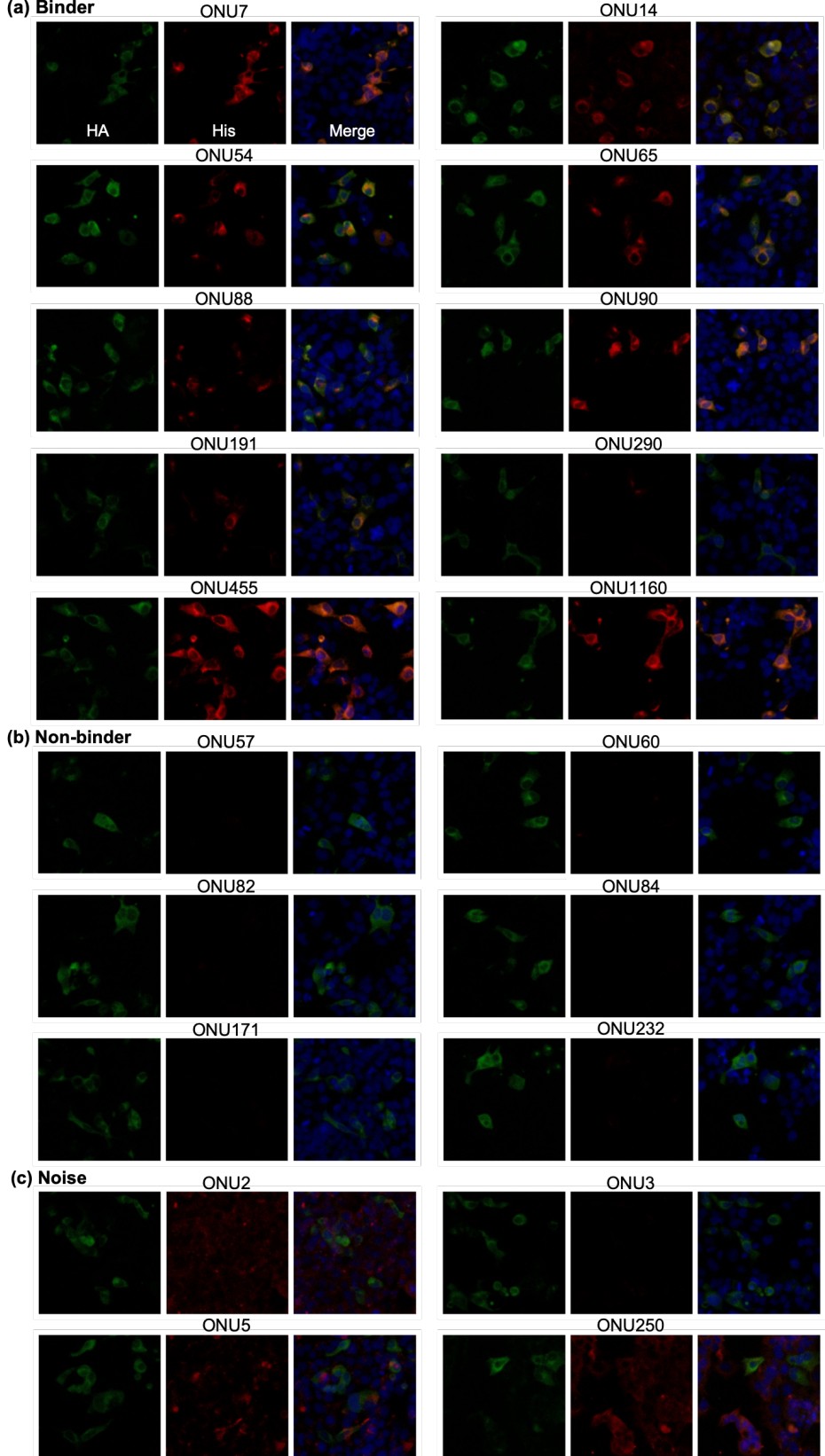

Figure 5: Immunofluorescence staining analysis using (a) 10 binder-labeled, (b) six non-binder-labeled, and (c) four noise-labeled VHHs. HA-tagged IL-6 proteins were introduced into the cells and stained with His-tagged VHHs. Subsequently, HA-tags were visualized in green and His-tags in red.

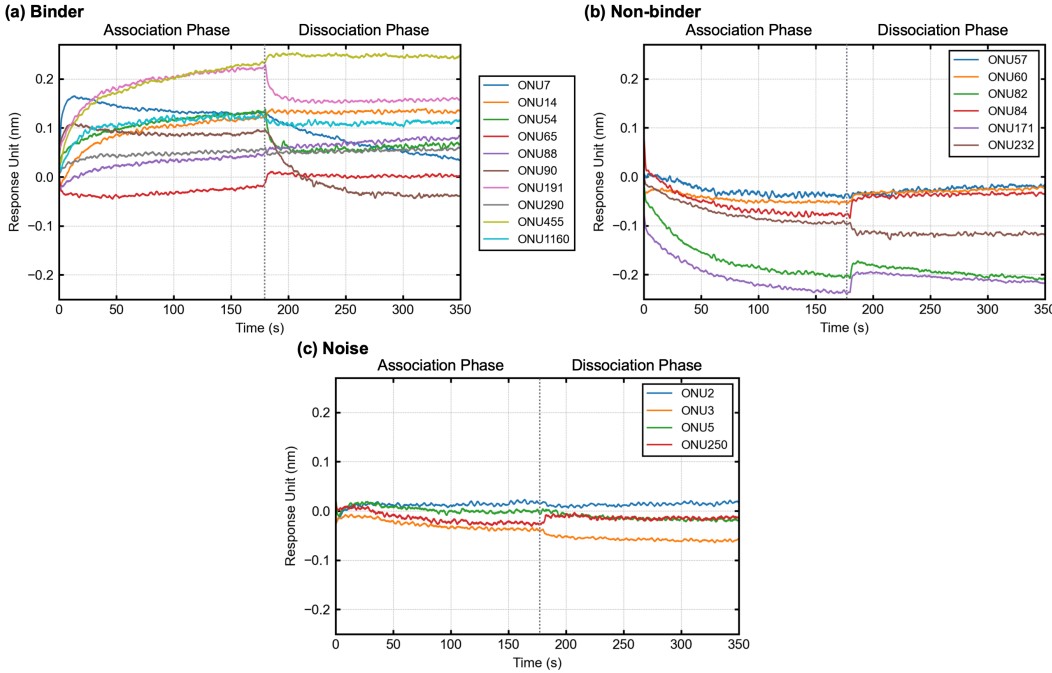

Figure 6: Biolayer interferometry analysis using (a) 10 binder-labeled, (b) six non-binder-labeled, and (c) four noise-labeled VHHs. Each line represents the association and dissociation curve of each VHH on the wild-type IL-6. When a line moves up from the base point in the association phase, the VHH is considered a binder.

protein synthesis system because of the limitations of the phage display method. Even if a protein of interest functioned as a fusion protein with the g3p protein on a phage, it was not always possible to express the protein alone in a soluble form with function [16]. However, if a sufficient signal was observed in the mother library, then the clones must have been truly present as heavy-chain antibodies, at least in the alpaca body. The remaining 10 clones all showed binding to the wild-type IL-6 protein, as shown in Figure 5(a). The immunofluorescence staining analysis showed strong to weak signals, which probably reflected avidity differences between the clones. Note that the calculated p-values did not correlate with the staining intensity by a simple inverse relationship. The biolayer interferometry (BLI) analysis revealed that the clones positively associated with the wild-type IL-6 protein with different association curves ($K_{on}$), dissociation curves ($K_{off}$), and KDs ($K_{off}/K_{on}$), as shown in Figure 6(a), although the sensitivity was relatively lower than that of the immunostaining analysis. All six non-binder-labeled clones showed negative results in both the immunostaining and BLI analyses, as shown in Figures 5(b) and 6(b), respectively. Of the five noise-labeled clones, one could not be isolated. The immunostaining analysis showed that the remaining four clones likely had nonspecific binding, as shown in Figure 5(c), but not to the wild-type IL-6 protein, as confirmed by the BLI analysis results shown in Figure 6(c). Accordingly, the sensitivity and specificity of our labeling method can be considered sufficiently high.

## A.4 Benchmarks

### A.4.1 Data Splitting

For a test set, we randomly selected 15 mutants: P42A, T48A, E51A, I57A, I60A, K69A, C78A, S81A, E87A, L120A, L126A, L129A, Q144A, D162A, and T165A. The remaining 15 mutants and the wild type were used for model training. First, we trained the models by using only the wild-type IL-6 protein and evaluated the model performance on the test set. Then, we randomly selected one mutant from the mutants not contained in the test set and added it to the training set to evaluate each model's predictive performance on the test set. By repeating this procedure, we tracked each model's predictive performance for unknown mutants contained only in the test set. Because the order of adding mutants to the training set affected the model performance, we ran the same experiment five

times in shuffled order, and we report the averaged results in section 4.3. Table 4 summarizes the order in which mutants were added and the number of samples in each set.

### A.4.2 Model Implementations

We adopted AbAgIntPre because it is a state-of-the-art model designed for the same task setting as ours of predicting interactions solely from antigen and antibody sequences. At present, fewer studies have focused on developing machine learning models for predicting antigen-antibody interactions based only on amino acid sequences, as compared to PPI. However, antigen-antibody interactions that ignore non-protein antigens can be considered a subset of PPI, meaning that models designed for PPI are also applicable to antigen-antibody interactions. Thus, we adopted PIPR as a representative neural-network-based model designed for PPI. In addition, we used MLP as a simpler, shallower neural network model than AbAgIntPre and PIPR. Lastly, we chose LR as a classical machine learning model other than neural networks. The implementations of all the benchmark models are available at `https://github.com/cognano/AVIDa-hIL6`.

- **AbAgIntPre** [20]. We used the implementation[3] that is provided by AbAgIntPre's developers and released under Apache License 2.0 for the composition of k-spaced amino acid pairs (CKSAAP) [11] encoding. The calculation was performed using k = 0, 1, 2, 3, thus yielding a 1600-dimensional vector for each amino acid sequence. We also used the original PyTorch [33] implementation released under Apache License 2.0 for the AbAgIntPre model. We used the model parameters reported in the original paper [20].

- **PIPR** [10]. We reimplemented PIPR by using PyTorch with reference to the original implementation[4] released under Apache License 2.0. We changed the number of RCNN units from five to three, while the other parameters were the same as in the original paper [10]. Each RCNN unit had a one-dimensional max pooling with a kernel size of three, which shortened the sequence length by a third. Our dataset's maximum sequence length is 218, and the application of five RCNN units would have resulted in a sequence length shorter than one; thus, we reduced the number of units. In addition, we used the pretrained embeddings published by PIPR's developers in their original implementation as "vec5_CTC.txt."

- **Multi-Layer Perceptron (MLP).** We implemented one-hot encoding and an MLP with one hidden layer of 512 neurons and the rectified linear unit (ReLU) activation function by using PyTorch. The one-hot vectors of the VHHs and IL-6 proteins were flattened and concatenated for input to the MLP. We used zero padding to match the dimensions, thus yielding 8000-dimensional vectors for each VHH and IL-6 protein pair.

- **Logistic Regression (LR).** We implemented LR by using scikit-learn [34]. We used the default settings of scikit-learn v1.2.2 as model parameters. As with the MLP, the one-hot vectors of the VHHs and IL-6 proteins were flattened and concatenated for input to the LR model. We used zero padding to match the dimensions, thus yielding 8000-dimensional vectors for each VHH and IL-6 protein pair.

### A.4.3 Model Training

All three neural network-based models were trained on one NVIDIA Tesla V100 GPU on Google Colaboratory. The models were trained for 100 epochs with a batch size of 256. We used the Adam optimizer with a fixed learning rate of 0.0001 and no weight decay. Also, we used the binary cross-entropy with sigmoid activation as the loss function. During the model training, the GPU memory consumptions for AbAgIntPre, PIPR, and MLP were approximately 1.4, 1.6, and 1.1 GB, respectively. When using 16 IL protein types for training, the training times for AbAgIntPre, PIPR, and MLP were approximately 0.5, 1, and 0.5 hours, respectively.

The LR model, with L2 penalization, was trained using one Intel(R) Xeon(R) CPU @ 2.20 GHz on Google Colaboratory. The solver was the LBFGS algorithm, with 1,000 as the maximum number of iterations for convergence. When using 16 IL protein types for training, the training time for LR was approximately 0.5 hours. More detailed training information is available at `https://github.com/cognano/AVIDa-hIL6`.

---

[3]AbAgIntPre: `https://github.com/emersON106/AbAgIntPre`
[4]PIPR: `https://github.com/muhaochen/seq_ppi`

Table 4: Details of the data splitting.

| Experiment | #Antigen types | Added antigen | #Samples Binder | Non-binder | Total |
|---|---|---|---|---|---|
| | 1 | Wild-type | 540 | 12,480 | 13,020 |
| | 2 | G63A | 1,074 | 39,629 | 40,703 |
| | 3 | F153A | 2,232 | 49,940 | 52,172 |
| | 4 | T117A | 2,967 | 64,508 | 67,475 |
| | 5 | Q45A | 3,799 | 80,574 | 84,373 |
| Run 1 | 6 | E108A | 4,424 | 99,729 | 104,153 |
| | 8 | C72A, F102A | 5,503 | 148,208 | 153,711 |
| | 10 | P93A, E138A | 7,249 | 178,334 | 185,583 |
| | 12 | D54A, S135A | 8,737 | 222,447 | 231,184 |
| | 14 | D168A, D99A | 9,452 | 257,265 | 266,717 |
| | 16 | L90A, G105A | 10,564 | 282,279 | 292,843 |
| | 1 | Wild-type | 540 | 12,480 | 13,020 |
| | 2 | E138A | 1,942 | 22,052 | 23,994 |
| | 3 | C72A | 2,406 | 47,106 | 49,512 |
| | 4 | L90A | 2,656 | 59,150 | 61,806 |
| | 5 | F153A | 3,814 | 69,461 | 73,275 |
| Run 2 | 6 | E108A | 4,439 | 88,616 | 93,055 |
| | 8 | D99A, G105A | 5,737 | 124,168 | 129,905 |
| | 10 | D168A, D54A | 6,509 | 165,155 | 171,664 |
| | 12 | G63A, P93A | 7,387 | 212,858 | 220,245 |
| | 14 | Q45A, S135A | 9,214 | 244,286 | 253,500 |
| | 16 | F102A, T117A | 10,564 | 282,279 | 292,843 |
| | 1 | Wild-type | 540 | 12,480 | 13,020 |
| | 2 | G63A | 1,074 | 39,629 | 40,703 |
| | 3 | L90A | 1,324 | 51,673 | 52,997 |
| | 4 | G105A | 2,186 | 64,643 | 66,829 |
| | 5 | F102A | 2,801 | 88,068 | 90,869 |
| Run 3 | 6 | Q45A | 3,633 | 104,134 | 107,767 |
| | 8 | D99A, D54A | 4,562 | 155,467 | 160,029 |
| | 10 | E138A, E108A | 6,589 | 184,194 | 190,783 |
| | 12 | S135A, C72A | 8,048 | 224,610 | 232,658 |
| | 14 | P93A, D168A | 8,671 | 257,400 | 266,071 |
| | 16 | F153A, T117A | 10,564 | 282,279 | 292,843 |
| | 1 | Wild-type | 540 | 12,480 | 13,020 |
| | 2 | F102A | 1,155 | 35,905 | 37,060 |
| | 3 | D99A | 1,591 | 58,487 | 60,078 |
| | 4 | L90A | 1,841 | 70,531 | 72,372 |
| | 5 | G105A | 2,703 | 83,501 | 86,204 |
| Run 4 | 6 | D54A | 3,196 | 112,252 | 115,448 |
| | 8 | E138A, P93A | 4,942 | 142,378 | 147,320 |
| | 10 | C72A, E108A | 6,031 | 186,587 | 192,618 |
| | 12 | F153A, G63A | 7,723 | 224,047 | 231,770 |
| | 14 | Q45A, T117A | 9,290 | 254,681 | 263,971 |
| | 16 | D168A, S135A | 10,564 | 282,279 | 292,843 |
| | 1 | Wild-type | 540 | 12,480 | 13,020 |
| | 2 | D99A | 976 | 35,062 | 36,038 |
| | 3 | E108A | 1,601 | 54,217 | 55,818 |
| | 4 | F102A | 2,216 | 77,642 | 79,858 |
| | 5 | G63A | 2,750 | 104,791 | 107,541 |
| Run 5 | 6 | D168A | 3,029 | 117,027 | 120,056 |
| | 8 | F153A, Q45A | 5,019 | 143,404 | 148,423 |
| | 10 | T117A, G105A | 6,616 | 170,942 | 177,558 |
| | 12 | D54A, C72A | 7,573 | 224,747 | 232,320 |
| | 14 | P93A, S135A | 8,912 | 260,663 | 269,575 |
| | 16 | L90A, E138A | 10,564 | 282,279 | 292,843 |

