# AVIDa-hIL6: A Large-Scale VHH Dataset Produced from an Immunized Alpaca for Predicting Antigen-Antibody Interactions

**Hirofumi Tsuruta**[1*], **Hiroyuki Yamazaki**[1*], **Ryota Maeda**[1*], **Ryotaro Tamura**[1],
**Jennifer N. Wei**[2], **Zelda Mariet**[2], **Poomarin Phloyphisut**[2], **Hidetoshi Shimokawa**[2],
**Joseph R. Ledsam**[2], **Lucy Colwell**[2], **Akihiro Imura**[1*]

[1]COGNANO Inc., [2]Google LLC

{tsuruta, yamazaki, maeda, ryotarotamura, akihiroimura}@cognano.co.jp
{weijennifer, zmariet, poomarin, simokawa, jledsam, lcolwell}@google.com

## 1 Datasheet

The following questions were copied from "Datasheets for Datasets" [2].

### 1.1 Motivation

**For what purpose was the dataset created? Was there a specific task in mind? Was there a specific gap that needed to be filled? Please provide a description.**

AVIDa-hIL6 was created to facilitate the development of methods for predicting antigen-antibody interactions based on amino acid sequence information. In particular, AVIDa-hIL6 can be used for the binary classification task of whether a given antigen-antibody pair binds because AVIDa-hIL6 has reliable binary labels.

**Who created this dataset (e.g., which team, research group) and on behalf of which entity (e.g., company, institution, organization)?**

COGNANO Inc.

**Who funded the creation of the dataset? (If there is an associated grant, please provide the name of the grantor and the grant name and number.)**

COGNANO Inc.

**Any other comments?**

No.

### 1.2 Composition

**What do the instances that comprise the dataset represent (e.g., documents, photos, people, countries)? Are there multiple types of instances (e.g., movies, users, and ratings; people and interactions between them; nodes and edges)? Please provide a description.**

Instances of AVIDa-hIL6 are antigen-variable domain of heavy chain of heavy chain antibody (VHH) pairs. AVIDa-hIL6 has the wild type and 30 mutants of the human interleukin-6 (IL-6) protein as antigens.

**How many instances are there in total (of each type, if appropriate)?**

---

*Equal contribution.

AVIDa-hIL6 contains 573,891 data samples, comprising 20,980 binding pairs and 552,911 non-binding pairs.

**Does the dataset contain all possible instances or is it a sample (not necessarily random) of instances from a larger set? If the dataset is a sample, then what is the larger set? Is the sample representative of the larger set (e.g., geographic coverage)? If so, please describe how this representativeness was validated/verified. If it is not representative of the larger set, please describe why not (e.g., to cover a more diverse range of instances, because instances were withheld or unavailable).**

AVIDa-hIL6 extracts from all possible antigen-VHH pairs, only those that can be determined to be binding or non-binding by our labeling method, as shown in Section 3.1.

**What data does each instance consist of? "Raw" data (e.g., unprocessed text or images)or features? In either case, please provide a description.**

Each instance has amino acid sequences of the human IL-6 protein and VHH.

**Is there a label or target associated with each instance? If so, please provide a description.**

Yes. Each instance has a binary label for binding or non-binding pair.

**Is any information missing from individual instances? If so, please provide a description, explaining why this information is missing (e.g., because it was unavailable). This does not include intentionally removed information, but might include, e.g., redacted text.**

No.

**Are relationships between individual instances made explicit (e.g., users' movie ratings, social network links)? If so, please describe how these relationships are made explicit.**

No.

**Are there recommended data splits (e.g., training, development/validation, testing)? If so, please provide a description of these splits, explaining the rationale behind them.**

Yes. We recommend splitting the dataset based on the type of IL-6 protein to predict the impact of antigen mutations on antibody binding.

**Are there any errors, sources of noise, or redundancies in the dataset? If so, please provide a description.**

Yes. The results of biological experiments always contain background noise, such as binding to contaminating proteins. Therefore, our labels may contain errors.

**Is the dataset self-contained, or does it link to or otherwise rely on external resources (e.g., websites, tweets, other datasets)? If it links to or relies on external resources, a) are there guarantees that they will exist, and remain constant, over time; b) are there official archival versions of the complete dataset (i.e., including the external resources as they existed at the time the dataset was created); c) are there any restrictions (e.g., licenses, fees) associated with any of the external resources that might apply to a future user? Please provide descriptions of all external resources and any restrictions associated with them, as well as links or other access points, as appropriate.**

AVIDa-hIL6 is self-contained.

**Does the dataset contain data that might be considered confidential (e.g., data that is protected by legal privilege or by doctor-patient confidentiality, data that includes the content of individuals' non-public communications)? If so, please provide a description.**

No.

**Does the dataset contain data that, if viewed directly, might be offensive, insulting, threatening, or might otherwise cause anxiety? If so, please describe why.**

No.

**Does the dataset relate to people? If not, you may skip the remaining questions in this section.**

No.

**Does the dataset identify any subpopulations (e.g., by age, gender)? If so, please describe how these subpopulations are identified and provide a description of their respective distributions within the dataset.**

N/A.

**Is it possible to identify individuals (i.e., one or more natural persons), either directly or indirectly (i.e., in combination with other data) from the dataset? If so, please describe how.**

N/A.

**Does the dataset contain data that might be considered sensitive in any way (e.g., data that reveals racial or ethnic origins, sexual orientations, religious beliefs, political opinions or union memberships, or locations; financial or health data; biometric or genetic data; forms of government identification, such as social security numbers; criminal history)? If so, please provide a description.**

N/A.

**Any other comments?**

No.

## 1.3 Collection Process

**How was the data associated with each instance acquired? Was the data directly observable (e.g., raw text, movie ratings), reported by subjects (e.g., survey responses), or indirectly inferred/derived from other data (e.g., part-of-speech tags, model-based guesses for age or language)? If data was reported by subjects or indirectly inferred/derived from other data, was the data validated/verified? If so, please describe how.**

We acquired data from a single alpaca immunized with IL-6 protein.

**What mechanisms or procedures were used to collect the data (e.g., hardware apparatus or sensor, manual human curation, software program, software API)? How were these mechanisms or procedures validated?**

We used procedures such as the immunization of an alpaca, the construction of phage libraries, affinity selection by biopanning, and sequence analysis by next-generation sequencing (NGS) to collect data. The detailed procedures are provided in Section 3.1 and Appendix A.2.

**If the dataset is a sample from a larger set, what was the sampling strategy (e.g., deterministic, probabilistic with specific sampling probabilities)?**

N/A.

**Who was involved in the data collection process (e.g., students, crowdworkers, contractors) and how were they compensated (e.g., how much were crowdworkers paid)?**

Five authors from COGNANO Inc. were involved in the data collection process.

**Over what timeframe was the data collected? Does this timeframe match the creation timeframe of the data associated with the instances (e.g., recent crawl of old news articles)? If not, please describe the timeframe in which the data associated with the instances was created.**

The released dataset was collected from March to April 2019.

**Were any ethical review processes conducted (e.g., by an institutional review board)? If so, please provide a description of these review processes, including the outcomes, as well as a link or other access point to any supporting documentation.**

All animal experiments on an alpaca were conducted in accordance with the KYODOKEN Institute for Animal Science Research and Development (Kyoto, Japan) and the ARRIVE (Animal Research: Reporting of *In Vivo* Experiments) guidelines[2]. Veterinarians performed breeding, health maintenance, and immunization by adhering to the published Guidelines for Proper Conduct of Animal Experiments

---

[2]ARRIVE guidelines: `https://arriveguidelines.org`

by the Science Council of Japan. The KYODOKEN Institutional Animal Care and Use Committee approved the protocols for these studies (KYODOKEN protocol number 20190216).

**Does the dataset relate to people? If not, you may skip the remaining questions in this section.**

No.

**Did you collect the data from the individuals in question directly, or obtain it via third parties or other sources (e.g., websites)?**

N/A.

**Were the individuals in question notified about the data collection? If so, please describe (or show with screenshots or other information) how notice was provided, and provide a link or other access point to, or otherwise reproduce, the exact language of the notification itself.**

N/A.

**Did the individuals in question consent to the collection and use of their data? If so, please describe (or show with screenshots or other information) how consent was requested and provided, and provide a link or other access point to, or otherwise reproduce, the exact language to which the individuals consented.**

N/A.

**If consent was obtained, were the consenting individuals provided with a mechanism to revoke their consent in the future or for certain uses? If so, please provide a description, as well as a link or other access point to the mechanism (if appropriate).**

N/A.

**Has an analysis of the potential impact of the dataset and its use on data subjects (e.g., a data protection impact analysis) been conducted? If so, please provide a description of this analysis, including the outcomes, as well as a link or other access point to any supporting documentation.**

N/A.

**Any other comments?**

No.

### 1.4 Preprocessing/cleaning/labeling

**Was any preprocessing/cleaning/labeling of the data done (e.g., discretization or bucketing, tokenization, part-of-speech tagging, SIFT feature extraction, removal of instances, processing of missing values)? If so, please provide a description. If not, you may skip the remainder of the questions in this section.**

Yes. We preprocessed sequences obtained from NGS as described in Appendix A.2 and labeled antigen-VHH pairs as described in Section 3.1.

**Was the "raw" data saved in addition to the preprocessed/cleaned/labeled data (e.g., to support unanticipated future uses)? If so, please provide a link or other access point to the "raw" data.**

Yes. The raw data are available at `https://avida-hil6.cognanous.com`.

**Is the software used to preprocess/clean/label the instances available? If so, please provide a link or other access point.**

Yes. The codes can be found at `https://github.com/cognano/AVIDa-hIL6`.

**Any other comments?**

No.

### 1.5 Uses

**Has the dataset been used for any tasks already? If so, please provide a description.**

In our paper, AVIDa-hIL6 was used to predict the interaction between the IL-6 protein and VHH using machine learning models.

**Is there a repository that links to any or all papers or systems that use the dataset? If so, please provide a link or other access point.**

No.

**What (other) tasks could the dataset be used for?**

We expect that AVIDa-hIL6 can be used to predict binding sites such as epitopes and paratopes. For more details, please refer to Section 5.1.

**Is there anything about the composition of the dataset or the way it was collected and preprocessed/cleaned/labeled that might impact future uses? For example, is there anything that a future user might need to know to avoid uses that could result in unfair treatment of individuals or groups (e.g., stereotyping, quality of service issues) or other undesirable harms (e.g., financial harms, legal risks) If so, please provide a description. Is there anything a future user could do to mitigate these undesirable harms?**

VHHs are known to have low toxicity to humans, and several VHH drugs have been approved to date [1]. As our dataset was derived from alpacas, even if the risk of autoimmune adverse events is low for alpacas, it may not be for humans. Therefore, a phase I clinical trial cannot be omitted for each clone for the time being. For more details, please refer to Section 5.2.

**Are there tasks for which the dataset should not be used? If so, please provide a description.**

AVIDa-hIL6 cannot be used for any tasks related to commercial drug development.

**Any other comments?**

No.

## 1.6 Distribution

**Will the dataset be distributed to third parties outside of the entity (e.g., company, institution, organization) on behalf of which the dataset was created? If so, please provide a description.**

Yes. AVIDa-hIL6 is publicly available.

**How will the dataset will be distributed (e.g., tarball on website, API, GitHub)? Does the dataset have a digital object identifier (DOI)?**

We released AVIDa-hIL6 on Zenodo with a DOI: `https://doi.org/10.5281/zenodo.7935862`.

**When will the dataset be distributed?**

We have already released AVIDa-hIL6.

**Will the dataset be distributed under a copyright or other intellectual property (IP) license, and/or under applicable terms of use (ToU)? If so, please describe this license and/or ToU, and provide a link or other access point to, or otherwise reproduce, any relevant licensing terms or ToU, as well as any fees associated with these restrictions.**

AVIDa-hIL6 is released under a CC BY-NC 4.0 license.

**Have any third parties imposed IP-based or other restrictions on the data associated with the instances? If so, please describe these restrictions, and provide a link or other access point to, or otherwise reproduce, any relevant licensing terms, as well as any fees associated with these restrictions.**

No.

**Do any export controls or other regulatory restrictions apply to the dataset or to individual instances? If so, please describe these restrictions, and provide a link or other access point to, or otherwise reproduce, any supporting documentation.**

No.

**Any other comments?**

No.

### 1.7 Maintenance

**Who is supporting/hosting/maintaining the dataset?**

The dataset and the code used to generate the dataset are hosted on Zenodo and GitHub, respectively, to ensure high availability and long-term preservation. A website at https://avida-hil6.cognanous.com, on a hosting service contracted by COGNANO Inc., provides a detailed description of the dataset and links to Zenodo and GitHub. This website will be maintained by us and COGNANO Inc.'s engineering team.

**How can the owner/curator/manager of the dataset be contacted (e.g., email address)?**

Please contact avid@cognano.co.jp.

**Is there an erratum? If so, please provide a link or other access point.**

There are no errata for our initial release. Errata will be published on the dataset website and GitHub when needed.

**Will the dataset be updated (e.g., to correct labeling errors, add new instances, delete instances')? If so, please describe how often, by whom, and how updates will be communicated to users (e.g., mailing list, GitHub)?**

If we find any issues with AVIDa-hIL6 and update it, we will release an updated version on Zenodo.

**If the dataset relates to people, are there applicable limits on the retention of the data associated with the instances (e.g., were individuals in question told that their data would be retained for a fixed period of time and then deleted)? If so, please describe these limits and explain how they will be enforced.**

N/A.

**Will older versions of the dataset continue to be supported/hosted/maintained? If so, please describe how. If not, please describe how its obsolescence will be communicated to users.**

Yes. If we plan to update the dataset, we will maintain the old version and then release the updated version.

**If others want to extend/augment/build on/contribute to the dataset, is there a mechanism for them to do so? If so, please provide a description. Will these contributions be validated/verified? If so, please describe how. If not, why not? Is there a process for communicating/distributing these contributions to other users? If so, please provide a description.**

We welcome and encourage others to extend/augment/build on/contribute to the dataset. If others would like to contribute to AVIDa-hIL6, they can submit a pull request on GitHub or contact us via email.

**Any other comments?**

No.

## 2 Statement of Responsibility

The authors bear all responsibility for violations of rights related to AVIDa-hIL6.