# OpenReview forum: "AVIDa-hIL6: A Large-Scale VHH Dataset Produced from an Immunized Alpaca for Predicting Antigen-Antibody Interactions"
_NeurIPS.cc/2023/Track/Datasets_and_Benchmarks — NeurIPS 2023 Datasets and Benchmarks Poster_

### Official Review · Reviewer_Bn86 · 2023-07-26

**Rating:** 7
**Confidence:** 2
**Correctness:** Yes
**Clarity:** Yes

**Strengths:**

- The data size is much larger than before
- The data is extracted from wet experiments

**Additional Feedback:**

N/A

**Documentation:**

Yes

**Ethics:**

No concern

**Limitations:**

Yes

**Opportunities For Improvement:**

- It will be even better if the authors can provide the complex structure. Even the users can predict a single protein structure by AlpaFold, predicting the structure of how antigen and antibody bind seems still not a trivial task.

**Relation To Prior Work:**

Yes

**Summary And Contributions:**

The authors propose AVIDa-hIL6, a novel dataset for antigen-antibody interactions. The dataset is created by immunizing alpacas and extract the binding information. The data provide both sequence of antigen and antibody. It is also evaluated on some existing methods.

---

> ### Author Response · Authors · 2023-08-18
> **Response to Reviewer Bn86**
>
> We thank the reviewer for the valuable comment that we respond to below.
>
> ---
>
> ### Opportunities For Improvement
>
> **Q1 It will be even better if the authors can provide the complex structure. Even the users can predict a single protein structure by AlpaFold, predicting the structure of how antigen and antibody bind seems still not a trivial task.**
>
> Thank you very much for your constructive feedback.
> Structure determination of antigen-antibody complexes is crucial in drug discovery.
> However, the required financial, time, and human resources made it virtually impossible to determine the structures of all antigen-VHH pairs in our large dataset.
> Furthermore, as you mentioned, prediction of an antigen-antibody complex's structure with AlphaFold is still a challenging task.
>
> Instead, we believe that the important next steps are acquisition of antigen-VHH interaction datasets for a larger number of antigens, indirect epitope and paratope determination through machine learning model development, and subsequent computational structure prediction.
> Then, it would be feasible to perform experimental structure determination for a small number of scientifically important antigen-VHH complexes to verify and improve the accuracy of structure prediction.

---

### Official Review · Reviewer_CS39 · 2023-07-28
**A Novel Dataset for Antibody Binding as a Promising Biomedical Benchmark using artificial mutations and ML**

**Rating:** 6
**Confidence:** 3

**Strengths:**


1. The proposed technology presents a novel way of generating antibody-binding data of significantly increased volume, which could enhance the development of therapeutic antibodies.

2. The dataset contains antibody sequences that are positive for some IL-6 mutants and negative for others, thus can provide important insights for understanding how antigen mutations affect antibody binding.

3. Additionally, the authors developed a noise reduction algorithm using the negative control to avoid false positives and improve label reliability.

4. The code and the data are truly easily accessible online, well documented, and clearly organized.


**Additional Feedback:**


This paper describes a novel technique for generating an antigen-antibody interactions dataset.
The experimental setting and the motivation for generating this dataset are clearly described and the dataset webpage with documentation is well-organized, accessible, and aesthetically beautiful. The authors also acknowledged that one of the main limitations is the lack of sample diversity: the dataset contains exactly one protein (IL-6) and its artificial mutants as an antigen. At the moment, the experiments allow for a very focused narrow application and would generalize for the analysis of antibody binding to unknown mutants of a known antigen.
In addition, the authors explicitly and rigorously reported the exact dimensions of the used datasets and model input. In line with expectations, the resulting classes and very unbalances (~3.7% out of 574 thousand data samples comprising binding pairs) present a computational challenge and call for deeper analysis in the future.
Overall, I believe that this technology for dataset creation can be impactful and scaled further to become a very important biomedical benchmark that could lead to major breakthroughs in drug discovery using ML.


**Clarity:**

The paper is generally well-written, providing clear explanations and presenting the main ideas coherently. However, some details could be further elaborated to enhance clarity and completeness. One area that could benefit from more information is the similarity analysis between samples in the test and train datasets. For instance, providing information on the closeness of artificially introduced alanines' locations and their impact on antibody binding would offer valuable insights into the model's generalizability. Additionally, the paper could benefit from more precise explanations regarding the sample collection process. Specifying the number of lymph nodes collected at each time point after immunization, and the intervals between immunizations. Addressing the discrepancy in the number of lymph node samples collected at the 2-day time point and explaining any data correction or balancing strategies employed to account for this variation would be valuable for a more comprehensive understanding of the data collection process. Furthermore, providing more context on whether NGS was conducted before targeting and its potential implications for further analysis would also be helpful. Lastly, while Section 4.2 mentions the total amount of compute and the type of resources used, additional details about the computational setup, hyperparameters, and any parallelization techniques employed would benefit researchers interested in replicating the experiments.

**Correctness:**

The work looks generally correct, however, I have the following specific questions.
Selection of 100 Sequences: The process of selecting 100 sequences from the 650 VHH sequences with binding to specific IL-6 protein types but non-binding to others should be clearly described in the submission. Could you please provide details on the criteria used for selection and whether these 100 sequences are representative of the overall dataset?

The similarity of 15 Mutants: If the submission mentions randomly selecting 15 mutants for the test set, it would be beneficial to clarify how similar these mutants are to each other and the training dataset. Additionally, information about the similarity metric used (e.g., sequence identity) would help assess the diversity of the selected mutants.

Purpose of Adding 1 Mutant to the Training Set: If the submission describes adding one mutant from the remaining 15 to the training set, could you please explain the rationale behind this specific choice? The purpose of this addition and its impact on model performance should be clearly justified.


**Documentation:**

The authors did a superb job creating accessible, well-organized, esthetically beautiful, and user-friendly interfaces with documentation, data, and code. I was able to quickly find and download the dataset (from https://avida-hil6.cognanous.com/) and the code with detailed descriptions and visualizations is openly published on GitHub (https://github.com/cognano/AVIDa-hIL6).
There has been clearly a significant effort toward creating reproducible and accessible research!

The dataset is released under the Creative Commons Attribution-NonCommercial 4.0 International License, and the code under the MIT License.
Could the maintenance plan be added?


**Ethics:**

The authors have reported that all animal experiments on an alpaca were conducted in accordance with the KYODOKEN Institute for Animal Science Research and Development (Kyoto, Japan); Veterinarians performed breeding, health maintenance, and immunization by adhering to the published Guidelines for Proper Conduct of Animal Experiments by the Science Council of Japan. And the KYODOKEN Institutional Animal Care and Use Committee approved the protocols for these studies (KYODOKEN protocol number 20190216). I assume it implies that the procedures were performed to be as tolerable as possible for the animals and minimize cruelty, correct? Could there anything else be done additionally? Are there any computational ways to optimize the process? Is there any additional information that could be helpful for other researchers when creating a similar dataset in this regard?

The authors also mentioned that a low risk of autoimmune adverse events is guaranteed for alpacas but not for humans. What does guarantee that?

**Limitations:**

The authors have acknowledged several limitations of their work, which is commendable. However, there are opportunities to provide more comprehensive discussions on some of these limitations and their potential societal impacts:
Addressing Imbalance in Labels: The strong imbalance in the dataset, with only about 3.7% of the data samples being binding pairs, impacts the performance of machine learning models. The authors could elaborate on the techniques used to handle this class imbalance, such as oversampling, undersampling, or using class weights, to prevent the models from being heavily biased toward the majority class.


Lack of Antigen Diversity: The dataset's narrow focus on a single protein (IL-6) as the antigen limits the applicability of the model to predict interactions with new emerging antigens. The authors could discuss potential strategies for increasing antigen diversity in future work, such as immunizing alpacas with multiple antigens or exploring other sources of data to incorporate a wider variety of antigens.


Consideration of Natural Mutations: While the authors introduced artificial mutations for the IL-6 protein, they also acknowledged that natural mutations can be more complex, involving simultaneous mutations at different sites. It would be beneficial to discuss the potential impact of this limitation on the model's ability to predict interactions with naturally occurring mutations.


Ethical and Societal Implications: Given the potential use of the dataset and machine learning models for drug discovery and antibody development, it is essential to address any potential negative societal impacts. For example, the authors could discuss in more detail the importance of ensuring ethical considerations in the use of animal models and the potential risks associated with the development and use of therapeutic antibodies.


Potential Bias in Data Generation: The method of generating the dataset using alpaca immunization may introduce biases in the dataset. The authors should acknowledge and discuss these potential biases and their implications on the generalizability of the model's predictions.


Addressing Computational Challenges: As the dataset size is substantial, the authors could elaborate on the computational challenges faced during model training and how they addressed these challenges, such as using distributed computing or parallelization techniques.
Additionally, could some basic models like LRs, and trees be used for comparison?


**Opportunities For Improvement:**

Opportunities for Improvement:

Explicit References for Empirical Determinations: To enhance transparency and credibility, the submission should provide explicit references or sources for the empirical determinations made by domain experts. For instance, clarify how the value of 10^(2.5) was determined in line 204, and whether there might be a typographical error missing the minus sign. Similarly, please specify the experts and criteria involved in the selection of VHH candidates for Immunofluorescence staining (line 247).
Could you please address why certain antibody structures might pose challenges for sequencing and elaborate on what specific structural features of VHHs make them more amenable to DNA sequencing compared to conventional antibodies?

Utilization of Non-Significant Samples: The statement that "non-significant" labels based on p-values exceeding 0.05 were not used for supervised learning raises the question of whether these samples can be utilized for validation or other purposes. Clarifying the potential uses of non-significant samples will help to better understand the dataset's completeness and potential applications.

Benchmarking Details: The submission could include more information about the benchmarking process, such as the selection criteria for the three neural network-based baseline models used. Providing insights into the choice of these models will enhance the reproducibility and interpretability of the results.


**Relation To Prior Work:**

Great and detailed comparison of the previously published antigen-antibody interactions datasets with thorough discussion!
Prior datasets lack explicit information on the antigen corresponding to each antibody. The authors used existing assets, cited the creators, and mentioned the license of the assets. Table 1 provides the numbers of samples for each of the five publicly accessible datasets and denotes relevant information that was missing or not available for part of the dataset. The presented work uses immunized alpacas to generate a large amount of VHH sequence data in comparison to other datasets. Moreover, a novel approach was proposed to address the challenge of the identification of binding and non-binding pairs to generate the data labels.

**Summary And Contributions:**

The authors created a new dataset called AVIDa-hIL6 for predicting antigen-antibody interactions, which contains amino acid sequences of antigens and antibodies and binary labels for binding and non-binding pairs. They have designed a novel data generation method, including data labeling, which can theoretically target any antigen using a live alpaca's immune system.

More specifically, 30 artificial mutations were introduced into the IL-6 protein and used as an antigen in addition to wild-type IL-6 to identify binding vs. non-binding protein-VHH pairs using their amino acid sequence.
Three types of neural network-based baseline models were implemented and used to benchmark the prediction of antigen-antibody interactions. As a result, they additionally confirmed experimentally that models capture the impact of antigen mutations on antibody binding.

---

> ### Author Response · Authors · 2023-08-18
> **Response to Reviewer CS39 (Part 1/5)**
>
> We thank the reviewer for careful reading and the insightful comments that we address below.
> We updated the paper accordingly, with the changes highlighted in red.
>
> ---
>
> ### Opportunities For Improvement
>
> **Q1 Explicit References for Empirical Determinations: To enhance transparency and credibility, the submission should provide explicit references or sources for the empirical determinations made by domain experts. For instance, clarify how the value of 10^(2.5) was determined in line 204, and whether there might be a typographical error missing the minus sign. Similarly, please specify the experts and criteria involved in the selection of VHH candidates for Immunofluorescence staining (line 247).**
>
> Thank you for this very important question.
> First, the value of 10^(2.5) was correct, but we needed to include important information in the denominator and numerator of the ratio of p-values.
> Thus, we added this information in "Step 5. Data Labeling" in Section 3.1 (lines 217-218).
> Also, the value of 10^(2.5) was determined by an author, who is a biologist, according to feedback from biological experiments in our previous studies [29].
> We also added this explanation in "Step 5. Data Labeling" (lines 219-220).
> Finally, we verified that the noise reduction algorithm using this empirical value works well, as explained in Section 3.3 and Appendix A.3.2.
>
> Next, we selected VHHs with suspect labels in order of the number of reads for label verification.
> Because this explanation was missing in the paper, we updated the first paragraph of Section 3.3 to clarify this selection criterion.
>
> [29] Maeda, R., Fujita, J., Konishi, Y., Kazuma, Y., Yamazaki, H., Anzai, I., Watanabe, T., Yamaguchi, K., Kasai, K., Nagata, K., et al.: A panel of nanobodies recognizing conserved hidden clefts of all SARS-CoV-2 spike variants including Omicron. Communications Biology 5, 669 (2022)
>
> ---
>
> **Q2 Could you please address why certain antibody structures might pose challenges for sequencing and elaborate on what specific structural features of VHHs make them more amenable to DNA sequencing compared to conventional antibodies?**
>
> Thank you for pointing out these important points in understanding the benefits of using VHHs.
> In the revised paper, we added a detailed explanation of why DNA sequencing of conventional antibodies is difficult and what structural features of VHHs make them more efficient to sequence.
> Please refer to the paragraph "Antibody Type" in Section 2.
>
> ---
>
> **Q3 Utilization of Non-Significant Samples: The statement that "non-significant" labels based on p-values exceeding 0.05 were not used for supervised learning raises the question of whether these samples can be utilized for validation or other purposes. Clarifying the potential uses of non-significant samples will help to better understand the dataset's completeness and potential applications.**
>
> Thank you for your constructive suggestions.
> Our dataset contains 1,998,127 non-significant-labeled samples, including 325,865 unique VHH sequences that are certainly present in the alpaca body.
> Although we did not use these samples in this work, they may be helpful in the future for pre-training via self-supervised learning, as used by existing antibody-specific language models [37,32,26].
> Leem et al. [26] demonstrated that embeddings from pre-trained language models reflect various biologically meaningful aspects and can be leveraged for various downstream tasks via transfer learning.
> To clarify this, we added an explanation of the potential use of non-significant-labeled samples to "Step 5. Data Labeling" in Section 3.1 (lines 206-209).
>
> [26] Leem, J., Mitchell, L.S., Farmery, J.H., Barton, J., Galson, J.D.: Deciphering the language of antibodies using self-supervised learning. Patterns 3(7), 100513 (2022)
>
> [32] Olsen, T.H., Moal, I.H., Deane, C.M.: AbLang: an antibody language model for completing antibody sequences. Bioinformatics Advances 2(1), vbac046 (2022)
>
> [37] Ruffolo, J.A., Gray, J.J., Sulam, J.: Deciphering antibody affinity maturation with language models and weakly supervised learning. arXiv preprint arXiv:2112.07782 (2021)

---

> ### Author Response · Authors · 2023-08-18
> **Response to Reviewer CS39 (Part 2/5)**
>
> **Q4 Benchmarking Details: The submission could include more information about the benchmarking process, such as the selection criteria for the three neural network-based baseline models used. Providing insights into the choice of these models will enhance the reproducibility and interpretability of the results.**
>
> Thank you for your useful suggestions.
> We adopted AbAgIntPre because it is a state-of-the-art model designed for the same task setting as ours of predicting interactions solely from antigen and antibody sequences.
> At present, fewer studies have focused on developing machine learning models for predicting antigen-antibody interactions based only on amino acid sequences, as compared to PPI.
> However, antigen-antibody interactions that ignore non-protein antigens can be considered a subset of PPI, meaning that models designed for PPI are also applicable to antigen-antibody interactions.
> Thus, we adopted PIPR as a representative neural-network-based model designed for PPI.
> In addition, we used MLP as a simpler, shallower neural network model than AbAgIntPre and PIPR.
> Lastly, we chose LR as a classical machine learning model other than neural networks.
> We added benchmark results for LR according to Q11.
>
> Because of the page limit, we added this explanation in Appendix A.4.2.
>
> ---
>
> ### Limitations
>
> **Q5 Addressing Imbalance in Labels: The strong imbalance in the dataset, with only about 3.7% of the data samples being binding pairs, impacts the performance of machine learning models. The authors could elaborate on the techniques used to handle this class imbalance, such as oversampling, undersampling, or using class weights, to prevent the models from being heavily biased toward the majority class.**
>
> Thank you for this important suggestion.
> As you mentioned, label imbalance in our dataset definitely impacted the model performance.
> Thus, techniques to address imbalanced labels, such as oversampling and undersampling, would be useful to improve the performance.
> Moreover, this discussion will be useful to other researchers who use our dataset in the future.
> We added this point in the first paragraph of Section 4.3 (lines 339-341).
>
> ---
>
> **Q6 Lack of Antigen Diversity: The dataset's narrow focus on a single protein (IL-6) as the antigen limits the applicability of the model to predict interactions with new emerging antigens. The authors could discuss potential strategies for increasing antigen diversity in future work, such as immunizing alpacas with multiple antigens or exploring other sources of data to incorporate a wider variety of antigens.**
>
> Thank you for your insightful suggestions.
> As discussed in Section 5.4, our potential strategy to overcome the lack of antigen diversity is to apply our data generation methods to a wider variety of antigens beyond the IL-6 protein.
> To ensure data quality, we immunized the alpaca with a single antigen at the same time.
> However, as you mentioned, combining AVIDa-hIL6 with other data sources would also be a valuable approach to improve the antigen diversity.
> Accordingly, we described this issue in the last part of the second paragraph of Section 5.4.
>
> ---
>
> **Q7 Consideration of Natural Mutations: While the authors introduced artificial mutations for the IL-6 protein, they also acknowledged that natural mutations can be more complex, involving simultaneous mutations at different sites. It would be beneficial to discuss the potential impact of this limitation on the model's ability to predict interactions with naturally occurring mutations.**
>
> Thank you for pointing this out.
> In general, natural mutants include those that have similar functions and structures even with different amino acid sequences and point mutations that cause loss or gain of the antigenic protein's function.
> Because simple artificial mutants have little chance of reproducing these rare properties, the efficiency of data collection for predicting antigen-antibody interactions of mutants with these properties is low.
> Accordingly, in the first paragraph of Section 5.4, we elaborated on what kinds of data are difficult to obtain with our data generation method using artificial mutants.

---

> ### Author Response · Authors · 2023-08-18
> **Response to Reviewer CS39 (Part 3/5)**
>
> **Q8 Ethical and Societal Implications: Given the potential use of the dataset and machine learning models for drug discovery and antibody development, it is essential to address any potential negative societal impacts. For example, the authors could discuss in more detail the importance of ensuring ethical considerations in the use of animal models and the potential risks associated with the development and use of therapeutic antibodies.**
>
> As you pointed out, we must consider and address the risk to the animals, because we generated our data by using an animal model that was immunized with a target antigen that could potentially harm the animal.
> We explained this point in the second paragraph of Appendix A.1.
> Additionally, we referenced it in the first paragraph of Section 3.1 to make it easier for the reader to find the ethics statement for animal experiments.
>
> ---
>
> **Q9 Potential Bias in Data Generation: The method of generating the dataset using alpaca immunization may introduce biases in the dataset. The authors should acknowledge and discuss these potential biases and their implications on the generalizability of the model's predictions.**
>
> Thank you for this very valuable feedback.
> As you pointed out, it is clear that our dataset potentially contains data biases derived from the timing and body site of the library collection and the specific alpaca used in the experiments.
> To reduce these data biases, it would be beneficial to collect samples at multiple times of immunization, from multiple individuals with different VHH gene sequences, and from multiple animals of different species.
>
> Because this information will be very important to other researchers who use our dataset in the future, we added a new Section 5.3 to discuss potential data biases, including their causes and how to address them.
>
> ---
>
> **Q10 Addressing Computational Challenges: As the dataset size is substantial, the authors could elaborate on the computational challenges faced during model training and how they addressed these challenges, such as using distributed computing or parallelization techniques.**
>
> Thank you for this suggestion.
> All of the neural-network-based models that we used could be trained on a single GPU in around an hour without using distributed computing or parallelization techniques.
> To clarify the required computation, we added the training time and GPU memory usage for each model in Appendix A.4.3.
>
> ---
>
> **Q11 Additionally, could some basic models like LRs, and trees be used for comparison?**
>
> We appreciate your suggestion.
> As you mentioned, it would be useful to add classical machine learning models other than neural networks to the baseline models for performance comparison.
> We thus added benchmark results using logistic regression (LR) to Figure 4 and Section 4.3, and a description of the model to Section 4.2 and Appendix A.4.2.
> Additionally, for experimental reproducibility, we added the code and instructions to run LR to https://github.com/cognano/AVIDa-hIL6.
>
> ---
>
> ### Correctness
>
> **Q12 Selection of 100 Sequences: The process of selecting 100 sequences from the 650 VHH sequences with binding to specific IL-6 protein types but non-binding to others should be clearly described in the submission. Could you please provide details on the criteria used for selection and whether these 100 sequences are representative of the overall dataset?**
>
> Thank you for pointing this out.
> We randomly sampled 100 sequences from 650 VHH sequences such that the extracted sequences were representative of the overall dataset.
> We added the word "randomly" to the corresponding sentence in the first paragraph of Section 3.2 (line 236).
>
> ---
>
> **Q13 The similarity of 15 Mutants: If the submission mentions randomly selecting 15 mutants for the test set, it would be beneficial to clarify how similar these mutants are to each other and the training dataset. Additionally, information about the similarity metric used (e.g., sequence identity) would help assess the diversity of the selected mutants.**
>
> Thank you for this important suggestion.
> As we used artificial point mutations, each mutant's sequence identity with respect to the wild type was the same, differing only in the position where the alanine was introduced.
> We explained this point in Section 4.1 (line 284-285).
>
> Additionally, we moved Table 3 in Appendix A.4.1 of the original paper to Section 4.1; it is now Table 2 in the revised paper.
> This table serves to clarify that the positions where the alanine was introduced were evenly distributed in the training and test sets.

---

> ### Author Response · Authors · 2023-08-18
> **Response to Reviewer CS39 (Part 4/5)**
>
> **Q14 Purpose of Adding 1 Mutant to the Training Set: If the submission describes adding one mutant from the remaining 15 to the training set, could you please explain the rationale behind this specific choice? The purpose of this addition and its impact on model performance should be clearly justified.**
>
> Thank you for this important question.
> Our experimental scenario assumes that antigen mutants emerge one after another to evade the immune system, as in the COVID-19 pandemic.
> The addition of one mutant to the training set meant that we had already observed that mutant and knew its amino acid sequence and binding information with VHH antibodies.
> In such a scenario, our experiments aimed to evaluate the model's performance in predicting antibody candidates that would bind to future emerging mutants according to the binding information of antigens that have already been observed.
> We explained this point in the last part of Section 4.1.
>
> ---
>
> ### Clarity
>
> **Q15 One area that could benefit from more information is the similarity analysis between samples in the test and train datasets. For instance, providing information on the closeness of artificially introduced alanines' locations and their impact on antibody binding would offer valuable insights into the model's generalizability.**
>
> Thank you for your feedback.
> Following Q13, we explained the similarity of the training and test sets in the revised paper.
>
> As you mentioned, there is a clear correlation between the closeness of mutations and antibody binding, as seen in Figure 2(b).
> When focusing on the same VHH sequence, i.e., one row, it can be seen that mutants with mutations at closer positions tend to have the same label.
> Accordingly, we explained this point in the last sentence of the first paragraph in Section 3.2 to provide this valuable insight to other researchers.
> Note that, to understand this correlation accurately, it is essential to know the closeness of mutations on an antigen's three-dimensional structure, which is not possible at this time.
>
> ---
>
> **Q16 Additionally, the paper could benefit from more precise explanations regarding the sample collection process. Specifying the number of lymph nodes collected at each time point after immunization, and the intervals between immunizations. Addressing the discrepancy in the number of lymph node samples collected at the 2-day time point and explaining any data correction or balancing strategies employed to account for this variation would be valuable for a more comprehensive understanding of the data collection process.**
>
> We provide the number of lymph nodes collected at each time point at https://avida-hil6.cognanous.com, and we referred to this website in the last sentence of "Step 1. Immunization" in Section 3.1 of the revised paper.
> The interval between immunizations was about two weeks, as mentioned in Section 3.1.
> However, we cannot disclose the exact dates of each immunization.
> In addition, at this time, we do not specifically address the discrepancy in the number of lymph nodes collected, and we treat them equally as one library.
>
> ---
>
> **Q17 Furthermore, providing more context on whether NGS was conducted before targeting and its potential implications for further analysis would also be helpful.**
>
> We assume that you mean a library obtained before immunization with the target antigen, called a naïve library.
> Naïve libraries are not stimulated by the target antigen and thus usually contain few antibodies that interact with it.
> Also, because the immune system is considered very fluid, the VHHs in naïve libraries vary greatly depending on when they were collected.
> Therefore, for statistical analysis of the avidity of a binding VHH for a target antigen, libraries should be compared before and after panning with the target antigen; moreover, the best reference libraries containing the maximum diversity of binding clones would be those after stimulation with the target antigen.
>
> ---
>
> **Q18 Lastly, while Section 4.2 mentions the total amount of compute and the type of resources used, additional details about the computational setup, hyperparameters, and any parallelization techniques employed would benefit researchers interested in replicating the experiments.**
>
> Thank you for your useful feedback.
> First, we performed all of our experiments entirely on Google Colaboratory without using any parallelization techniques.
> We thus clarified in the last paragraph of Section 4.2 that we used Google Colaboratory as a computational platform.
>
> Moreover, we added a further description of the model parameters in Appendix A.4.2 and a description of the training parameters in a new Appendix A.4.3.
> We also referred to these two appendices in the last paragraph of Section 4.2 to make them easier for readers to find.

---

> ### Author Response · Authors · 2023-08-18
> **Response to Reviewer CS39 (Part 5/5)**
>
> ### Documentation
>
> **Q19 Could the maintenance plan be added?**
>
> The dataset and the code used to generate the dataset are hosted on Zenodo and GitHub, respectively, to ensure high availability and long-term preservation.
> A website at https://avida-hil6.cognanous.com, on a hosting service contracted by COGNANO Inc., provides a detailed description of the dataset and links to Zenodo and GitHub.
> This website will be maintained by us and COGNANO Inc.'s engineering team.
> Additionally, there are no errata for our initial release, but errata will be published on the dataset website and GitHub when needed.
>
> We explained these points in "1.7 Maintenance" in the supplementary materials.
>
> ---
>
> ### Ethics
>
> **Q20 The authors have reported that all animal experiments on an alpaca were conducted in accordance with the KYODOKEN Institute for Animal Science Research and Development (Kyoto, Japan); Veterinarians performed breeding, health maintenance, and immunization by adhering to the published Guidelines for Proper Conduct of Animal Experiments by the Science Council of Japan. And the KYODOKEN Institutional Animal Care and Use Committee approved the protocols for these studies (KYODOKEN protocol number 20190216). I assume it implies that the procedures were performed to be as tolerable as possible for the animals and minimize cruelty, correct? Could there anything else be done additionally? Are there any computational ways to optimize the process? Is there any additional information that could be helpful for other researchers when creating a similar dataset in this regard?**
>
> Your understanding is correct.
> To provide additional information, we explained in Appendix A.1 that we performed all animal experiments in accordance with the ARRIVE guidelines (https://arriveguidelines.org).
> Additionally, to help readers find the ethics statement for animal experiments, we referenced it in the first paragraph of Section 3.1.
> We expect that feedback from computational analyses would enable optimization of the immunization period and the number of samples, but methods for such optimization are still under consideration.
>
> ---
>
> **Q21 The authors also mentioned that a low risk of autoimmune adverse events is guaranteed for alpacas but not for humans. What does guarantee that?**
>
> We appreciate your constructive question.
> In general, antibody genes are activated in B lymphocytes, and their complementarity-determining regions coding paratopes are the only genes in which unrestricted mutations occur in vivo.
> Hence, antibodies that happen to bind to their own tissues can be produced.
> Mammals have an immune tolerance system that minimizes the production of such autoreactive antibodies through several mechanisms [19].
> Therefore, as our dataset was derived from alpacas, even if the risk of autoimmune adverse events is low for alpacas, it may not be for humans.
>
> We explained this point in Section 5.2.
>
> [19] Goodnow, C.C., Sprent, J., de St Groth, B.F., Vinuesa, C.G.: Cellular and genetic mechanisms of self tolerance and autoimmunity. Nature 435(7042), 590–597 (2005)

---

### Official Review · Reviewer_E2ji · 2023-07-31
**A nice benchmark work on Antigen-Antibody Interactions.**

**Rating:** 7
**Confidence:** 3
**Clarity:** The paper is well-written and easy to…

**Strengths:**

- This paper Introduces AVIDa-hIL6, the most extensive dataset for predicting antigen-antibody interactions, surpassing any other public dataset by tenfold. This dataset contains amino acid sequences of both antigens and antibodies, along with binary labels indicating binding and non-binding pairs. The dataset is of significance to the community.
- AVIDa-hIL6 encompasses the wild type and 30 mutants of the IL-6 protein as antigens, offering numerous sensitive cases where point mutations in IL-6 either enhance or inhibit antibody binding.
- The authors have devised a groundbreaking data generation method, including data labeling, by leveraging the immune system of a live alpaca. This innovative approach can be applied to any target antigen, in addition to IL-6.
- The authors also present benchmark results for predicting antigen-antibody interactions using neural network-based baseline models. These results validate that AVIDa-hIL6 serves as an invaluable benchmark for assessing a model's capability to capture the impact of antigen mutations on antibody binding.








**Additional Feedback:**

It would be great if the author could provide more background knowledge about antibodies and antigens, e.g., phage library.

**Correctness:**

It is technically sound, including data, benchmarks, and evaluation, with detailed documentation.

**Documentation:**

Yes. The documentation looks good. https://cognanous.com/datasets/avida-hil6 The website is good-looking.

**Ethics:**

The paper does not have ethical concerns.

**Limitations:**

It would be great if the authors could incorporate more machine learning models into the experiment, especially the state-of-the-art deep learning methods, e.g., transformers. MLP is a little out-of-date in this area.

**Opportunities For Improvement:**

It would be great if the authors could incorporate more machine learning models into the experiment, especially the state-of-the-art deep learning methods, e.g., transformer. MLP is a little out-of-date in this area.

**Relation To Prior Work:**

- This paper Introduces AVIDa-hIL6, the most extensive dataset for predicting antigen-antibody interactions, surpassing any other public dataset by tenfold.
- The author should provide more background knowledge about antibodies and antigens, e.g., phage library.

**Summary And Contributions:**

The authors have created AVIDa-hIL6, an extensive dataset for predicting antigen-antibody interactions in the variable domain of heavy chain antibodies (VHHs), obtained from an alpaca immunized with the human interleukin-6 (IL-6) protein as antigens. The dataset consists of 573,891 antigen-VHH pairs with full-length amino acid sequences, leveraging the simple structure of VHHs, which enables easy identification through DNA sequencing technology. Each antigen-VHH pair has reliable labels indicating binding or non-binding, generated using a novel labeling method. Additionally, AVIDa-hIL6 includes 30 different mutants of the wild-type IL-6 protein, introduced through artificial mutations. The dataset is of significance to the community.

---

> ### Author Response · Authors · 2023-08-18
> **Response to Reviewer E2ji**
>
> We thank the reviewer for these valuable comments that we address below.
> We updated the paper accordingly, with the changes highlighted in red.
>
> ---
>
> ### Opportunities For Improvement
>
> **Q1 It would be great if the authors could incorporate more machine learning models into the experiment, especially the state-of-the-art deep learning methods, e.g., transformer. MLP is a little out-of-date in this area.**
>
> We appreciate your constructive feedback.
> First, following reviewer CS39's Q11, we added benchmark results for logistic regression to Figure 4 and Section 4.3, although it is not a state-of-the-art model.
> Second, while we agree that MLP is out of date, we adopted it for comparison as a simpler, shallower neural network model than AbAgIntPre and PIPR, as described in Section 4.2 and Appendix A.4.2.
> Also, AbAgIntPre, proposed in 2022, is a state-of-the-art deep learning model in this area, including optimal sequence encoding.
> Finally, to our knowledge, there is not yet a transformer-based model that predicts antigen-antibody interactions based only on amino acid sequences.
> Hence, further research on model architectures dedicated to predicting antigen-antibody interactions is needed, as discussed in the last sentence of Section 4.3.
>
> ---
>
> ### Relation To Prior Work
>
> **Q2 The author should provide more background knowledge about antibodies and antigens, e.g., phage library.**
>
> Following your feedback, we added certain background knowledge on phage display and antigen-antibody interactions in the revised paper.
> First, we provided common uses of phage display with an additional reference [7] in "Step 2. Phage Library Construction" in Section 3.1.
> Second, we added a more detailed explanation of antigen-antibody interactions in Section 5.1.
>
> [7] Bazan, J., Całkosinski, I., Gamian, A.: Phage display—a powerful technique for immunotherapy: 1. introduction and potential of therapeutic applications. Human vaccines & immunotherapeutics 8(12), 1817–1828 (2012)

---

### Decision · Program_Chairs · 2023-09-22

**Decision:**

Accept (Poster)

**Comment:**

Overall Assessment:
The benchmark paper introduces AVIDa-hIL6, a groundbreaking and expansive dataset for predicting antigen-antibody interactions in the VHHs (variable domain of heavy chain antibodies) using human interleukin-6 (IL-6) proteins as antigens in immunized alpacas. The dataset sets a new standard in the field, offering 573,891 labeled antigen-VHH pairs, 30 different IL-6 mutants, and novel data labeling methods. The paper is highly significant for its innovative approach and its potential to accelerate advancements in therapeutic antibody development.

Strengths:
Extensive Dataset: AVIDa-hIL6 is the most extensive dataset of its kind, exceeding existing public datasets by tenfold.

Novel Data Generation Method: The authors leveraged a live alpaca's immune system to create a groundbreaking data generation and labeling method, with broad applicability beyond IL-6.

Mutant Variability: The dataset includes 30 different mutants of the IL-6 protein, allowing nuanced understanding of the effects of mutations on antibody binding.

Benchmark Models: Benchmark results are provided using neural network-based models, validating the dataset as a valuable resource for the community.

Documentation and Accessibility: The code and dataset are well-documented and easily accessible online, making the research highly reproducible.

Opportunities for Improvement:
Machine Learning Models: The paper would benefit from incorporating more modern machine learning methods, especially transformer models, as the current MLP models are considered outdated.

Empirical Determinations: Clarity could be enhanced by providing explicit references for empirical values and decisions.

Benchmarking Details: More information about the selection criteria for baseline models would improve the paper’s interpretability.

Antigen Diversity: The dataset focuses only on IL-6, limiting its applicability to other antigens. Future work could aim to include more antigen types.

Limitations:
Class Imbalance: With only 3.7% of the data samples being binding pairs, the class imbalance could impact machine learning model performance.

Computational Challenges: Given the dataset’s size, elaboration on computational methods employed during model training would be valuable.